



# Development of a national 7-day ensemble streamflow forecasting service for Australia

Hapu A. Prasantha Hapuarachchi[1], Mohammed A. Bari[2], Aynul Kabir[1], Mohammad M. Hasan[3], Fitsum
M. Woldemeskel[1], Nilantha Gamage[1], Patrick Sunter[1], Xiaoyong S. Zhang[1], David E. Robertson[4], James
C. Bennett[4], Paul M. Feikema[1]

[1]Bureau of Meteorology, 700 Collins Street, Docklands, VIC 3008, Australia
[2]Bureau of Meteorology, 1 Ord Street, West Perth, WA 6005, Australia
[3]Bureau of Meteorology, The Treasury Building, Parkes Place West, Canberra, ACT 2600, Australia
[4]Commonwealth Scientific and Industrial Research Organization, Research Way, Clayton, VIC 3168, Australia

*Correspondence to*: Hapu A. Prasantha Hapuarachchi (prasantha.hapuarachchi@bom.gov.au)

**Abstract.**

Reliable streamflow forecasts with associated uncertainty estimates are essential to manage and make better use of Australia's
scarce surface water resources. Here we present the development of an operational 7-day ensemble streamflow forecasting
service for Australia to meet the growing needs of users, primarily water and river managers, for probabilistic forecasts to
support their decision making. We test the modelling methodology for 100 catchments to learn the characteristics of different
rainfall forecasts from Numerical Weather Prediction (NWP) models, the effect of statistical processing on streamflow
forecasts, the optimal ensemble size, and parameters of a bootstrapping technique for calculating forecast skill. A conceptual
hourly rainfall-runoff model, GR4H (hourly) and lag and route channel routing model that are in-built in the Short-term Water
Information Forecasting Tools (SWIFT) hydrologic modelling package are used to simulate streamflow from input rainfall
and potential evaporation. The statistical Catchment Hydrologic Pre-Processor (CHyPP) is used for calibrating rainfall
forecasts, and the Error Reduction and Representation In Stages (ERRIS) model is used to reduce hydrological errors and
quantify hydrological uncertainty. Calibrating raw forecast rainfall with CHyPP is an efficient method to significantly reduce
bias and improve reliability for up to 7 lead days. We demonstrate that ERRIS significantly improves forecast skill up to 7
lead days. Forecast skills are highest in temperate perennially flowing rivers, while it is lowest in intermittently flowing rivers.
A sensitivity analysis for optimising the number of streamflow ensemble members for the operational service shows that more
than 200 members are needed to represent the forecast uncertainty. We show that the bootstrapping block size is sensitive to
the forecast skill calculation a bootstrapping block size of one month is recommended to capture maximum possible
uncertainty. We present benchmark criteria for accepting forecast locations for the public service. Based on the criteria, 209
forecast locations out of a possible 281 are selected in different hydro-climatic regions across Australia for the public service.
The service, which has been operational since 2019, provides graphical and tabular products of ensemble streamflow forecasts
along with performance information, for up to 7 lead days with daily updates.



## 1 Introduction

Optimal management of water resources requires support from accurate, reliable, and timely streamflow forecasts to make decisions. Practical and scientific benefits of predictive modelling of hydrological processes are evident (Shmueli, 2010) and have long been recognised. Water forecasting models can make significant contributions to drought mitigation and alleviation, optimal management of urban and agricultural water allocations, basin planning, hydropower generation, and flood management and mitigation (Buizer et al., 2016). Skilful streamflow forecasts can significantly contribute to improving

reservoir operation, water supply storage reliability, and environmental allocation (Delaney et al., 2020). In the long term, these predictive hydrological models can potentially bring enormous benefits to the environment and society, ensuring economic growth and environmental sustainability (Talukder and Hipel, 2020).

Water and flood managers need accurate streamflow forecast information at the longest possible lead time to make optimal water management decisions. There is a wide range of modelling techniques, from conceptual, physically-based, statistical,

and stochastic time series to modern hybrid artificial intelligence (AI) models that can be used for streamflow forecasting. Conceptual and physically-based models are more commonly used for short- and medium-term streamflow forecasting. Statistical models such as the Bayesian Joint Probability (BJP) model (Robertson and Wang, 2012; Zhao et al., 2016; Charles et al., 2018) are mostly used for monthly or seasonal time scales streamflow forecasting. Recently, machine learning tools based on data pre-processing techniques and swarm intelligence algorithms have been successfully used for short-term

streamflow forecasting (Niu et al., 2020). Commonly, rainfall forecasts from a Numerical Weather Prediction (NWP) model are used as input to a calibrated hydrological model for streamflow forecasting. Over the last few decades, NWP has moved from deterministic to probabilistic (ensemble) forecasting. As a result, probabilistic streamflow forecasting has become increasingly popular across the globe (Pappenberger et al., 2016; Roy et al., 2017; Wu et al., 2020). Probabilistic forecasts provide estimates of uncertainty involved in the forecasts that assist users in making informed decisions from the different

scenarios available.

There are several large-scale (continental and global) hydrological models run by communities around the world (Bierkens et al., 2015; Emerton et al., 2016). The Global Flood Awareness System (GloFAS) is one such forecasting system that can skilfully predict extreme events in large river basins up to 1 month ahead (Alfieri et al., 2013). The European Flood Awareness System (EFAS), operational since 2012, is a European Commission initiative developed by the Joint Research Centre (JRC)

for riverine flood preparedness across Europe. The service aims to provide harmonised early warnings and hydrological information to national agencies across Europe. The U.S. Hydrologic Ensemble Forecast Service (HEFS), run by the National Weather Services (NWS), provides ensemble streamflow forecasts that seamlessly span lead times from less than 1 hour up to several years and that are spatially and temporally consistent for river basins across the U.S. (Demargne et al., 2014). Siddique and Mejia (2017) report that the ensemble streamflow forecasts in the U.S. mid-Atlantic region remain skilful for lead times

up to 7 days. Post-processing of the forecasts increased forecast skills across lead times and spatial scales.





Australia is a land of extremes from droughts to floods and raging fires. It has a wide range of geographical and topographical features with a large central arid or semi-arid zone. The southeast and southwest regions are temperate, and the north has a tropical climate (Stern et al., 2000). These unique geographical features result in the most significant inter-annual variability of streamflow, floods, and droughts compared with other continents (Poff et al., 2006). During 2001-2009 south-eastern

Australia experienced its most severe drought since 1901, known as the 'Millennium Drought' (http://www.bom.gov.au/climate/drought/knowledge-centre/previous-droughts.shtml). The region had the most extended period of below-median rainfall and, as a result, inflows to major reservoirs were very low (Van Dijk et al., 2013). In particular, inflow to reservoirs located within the Murray-Darling River Basin, Australia's food bowl, was 50% of the previously recorded minimum. The drought had wide, long-lasting societal, economic and environmental impacts (Bureau of Meteorology, 2021).

As a result, the federal government passed the Water Act 2007 (https://www.legislation.gov.au/Details/C2017C00151) to implement a water security plan for the nation. One of the critical components of implementing the water security plan was developing and operationalising streamflow forecasting services at different temporal scales with special emphasis on short-term (hours to days) and medium-term (months to seasons) forecasts. The Bureau of Meteorology (BoM) launched a seasonal streamflow forecasting service (Woldemeskel et al., 2018; Feikema et al., 2018) in 2010

(http://www.bom.gov.au/water/ssf/history.shtml). A 7-day deterministic streamflow forecasting service (Hapuarachchi et al., 2016) was progressively developed during 2010-2017 and released to the public. More recently, stakeholders showed greater interest in probabilistic streamflow forecasts as it provides information on the uncertainty involved in the forecasts and supports users in making an informed decision with associated uncertainties. In response, the BoM launched the 7-day ensemble streamflow forecasting (SDF) service (http://www.bom.gov.au/water/7daystreamflow/) in December 2019, upgrading the

existing deterministic service. The upgraded service provides a set of forecasts to give an indication of a range of possible streamflow outcomes based on input forecast rainfall uncertainties for up to 7 days lead-time at an hourly scale at different river gauge stations with useful skill and reliability.

This paper describes the development of a SDF service, including the characteristics of different Numerical Weather Prediction (NWP) model rainfall forecasts, application of calibration to forecast rainfall, error modelling of streamflow forecasts, the

optimal ensemble size required to represent uncertainty band, parameters of a bootstrapping technique for calculating forecast skill, operational implementation of the service, and future work. The next sections of the paper describe the methodology, verification metrics, catchments and data, results, and discussion and future work.

## 2 Modelling methodology

The adopted hybrid dynamical-statistical streamflow forecasting method consists of several components – NWP calibration,

hydrological modelling and hydrological error modelling and we firstly introduce the components of the system. The forecasting system is premised on separating rainfall forecasting from hydrological modelling. This includes separating the



estimation of uncertainty in rainfall forecasts from the estimation of uncertainty in hydrological model. The system is thus a hybrid dynamical-statistical forecasting system. This setup has several key benefits: it makes the system highly modular, allowing new models (e.g., new NWPs) to be substituted into the system without the need to revise other components (e.g.,

the hydrological model). Second, it means that more appropriate techniques can be applied to estimate forecast uncertainty in each case: for example, error models are better able to handle the strong autocorrelation in streamflow than statistical calibration methods typically applied to rainfall forecasts.

Operationalisation of the system requires many practical scientific questions to be addressed. In this paper we seek to identify:

(a) the minimum ensemble size that can be used while maintaining robust performance

(b) how best to describe forecast skill when only limited hindcast dataset is available.

Our methods then describe the approach taken to answer these questions. A verification strategy is critical to answer the operationalisation questions and also provide an assessment of the forecast performance. We finally describe the verification strategy adopted for this study.

### 2.1 Calibration and evaluation of rainfall forecasts

Three NWP rainfall forecast products (Table 1) are evaluated for 100 catchments (at the outlets) in this study to understand their characteristics and to explore the impact of calibration. These are the European Centre for Medium-Range Weather Forecasts (ECMWF) atmospheric model ensemble forecasts (Richardson, 2000), Australian Community Climate and Earth-System Simulator-Global Ensemble (ACCESS-GE2) forecasts (O'Kane et al., 2008), and the BoM's Poor Man's Ensemble (PME) (Ebert, 2001), the ensemble mean of NWP models from Australia, UK, USA, Canada, Europe, and Japan. Note that

ACCESS-GE2 was a pre-operational product made available for this study, and a newer version (ACCESS-GE3) is now available. The average areal rainfall of each sub-catchment per each ensemble member is calculated by taking the area-weighted average of gridded forecast rainfall for all grid cells intersecting the catchment. The average forecast rainfall is post-processed using the Catchment Hydrologic Pre-processor (CHyPP) model (Robertson et al., 2013), which is based on a Bayesian Joint Probability (BJP) model that defines a spatially variable probabilistic relationship between NWP model forecast

rainfall and observed rainfall. The BJP model relates forecast rainfall to corresponding observations using a *log-sinh* transformed bivariate normal distribution. The *log-sinh* transformation is applied to normalise observed and forecast rainfall data and to homogenise its variance. The Schaake shuffle (Clark et al., 2004) method used in the CHyPP generates spatially and temporally coherent calibrated forecasts by linking samples from forecast probability distributions at each consecutive lead-time within the entire hindcast period for each forecast location within the catchment.

A leave-one-month-out cross-validation procedure (Hapuarachchi et al., 2016) is applied to calibrate and validate the CHyPP model for the data period of 36 months from 2014 to 2016. ACCESS-GE2 hindcast data is limited and 2014-2016 is the common period of data available for the selected NWP rainfall products. Given that PME is a merged post-processed product of many global NWP products, it shows negligible improvement when CHyPP is used on it (Shrestha et al., 2020). Therefore,





the PME forecasts are not post-processed. We also call a raw super-ensemble, a merged product of ECMWF, ACCESS-GE2,
and PME with 75 members (Table 1). By calibrating the forecasts with CHyPP, we generate *x* number of bias-corrected
statistically reliable ensemble members for each rainfall forecast product, ACCESS-GE2 and ECMWF and concatenate them
with PME (total *OES* members) that is referred to as a post-processed super-ensemble. The *x* to be determined based on the
analysis on optimum ensemble size (present below). Raw and post-processed rainfall forecasts are evaluated independently
(Table 2) for different lead times at the catchment scale for bias, precision, and reliability (see section 2.5 for details). Note
that PME is included here mainly for generating a deterministic streamflow forecast to embed in the ensemble plume of the
forecast products of the operational service.

**Table 1.** Raw rainfall forecast products

| Product | Lead-time (Hours) | Ensemble members | Spatial Resolution (km) | Temporal Resolution (Hours) |
|---|---|---|---|---|
| ECMWF | 360 | 50 | 20 | 3 (0 – 144 lead), 6 (144 – 360 lead) |
| PME | 228 | 1 | 50 | 3 |
| ACCESS-GE2 | 240 | 24 | 60 | 3 |
| Super-ensemble | 240 | 75 | Areal average | 1 |

### 2.2 Generating and evaluating streamflow forecasts

#### 2.2.1 Rainfall-runoff model and channel routing

The core hydrologic modelling package used here is the Short-term Water Information Forecasting Tools (SWIFT, Perraud et
al., 2015). SWIFT consists of many hydrologic modelling tools including conceptual hydrologic models, catchment routing
models, channel routing models, streamflow error models, and parameter optimization methods. It supports deterministic and
ensemble hydrologic modelling for the retrospective evaluation of catchment models using hindcast data and real-time
forecasting. Previous research conducted in Australia (Perrin et al., 2003; Coron et al., 2012; Van Esse et al., 2013; Bennett et
al., 2016; Kunnath-Poovakka and Eldho, 2019) as well as elsewhere have shown that GR4J (Perrin et al., 2003) and its variants
perform at least as well as other conceptual models in a range of environments at daily and hourly time-steps. Therefore, the
GR4H rainfall-runoff model (Bennett et al., 2014), an hourly variant of the daily GR4J model and lag and route channel routing
is implemented here. A nationally consistent flow direction map from the Australian Hydrological Geospatial Fabric
(Geofabric, Atkinson et al., 2008) is used to delineate each catchment into sub-catchments and sub-areas of 100-500 km$^2$ to
represent a semi-distributed model structure. The number of sub-areas varies for each catchment and depends on catchment
size and availability of gauging locations. A collection of sub-areas makes a sub-catchment where a streamflow gauge exists
at the outlet. GR4H is applied to each sub-area. Runoff generated in each sub-area is routed to the catchment outlet using the



lag and route method. The model parameters are calibrated for each sub-catchment using the Shuffle Complex Evolution-
University of Arizona (SCE-UA) algorithm (Duan et al., 1994) within the SWIFT package.

### 2.2.2 Hydrological error modelling

In addition to errors contributing to streamflow forecasts from observed and forecast rainfall (see section 3.2), there are errors
in both hydrological model structure and in calibrated model parameters. For an operational forecasting service, it is essential
to reduce the forecast uncertainty due to these errors as much as possible to provide highly reliable and accurate forecasts to
users. Using the Error Representation and Reduction In Stages (ERRIS) (Bennett et al., 2021; Li et al., 2021) method, we
explore the impact of error modelling on streamflow forecasts. ERRIS is applied to address different statistical properties of
the forecast error in four stages: i) hydrological model forecast and data normalisation, ii) non-linear bias correction, iii)
restricted autoregressive (AR) model updating, and iv) adjustment of residual distribution. After the hydrologic model and
routing model parameters are calibrated, the ERRIS parameters are calibrated for each sub-catchment from upstream to
downstream. In simulation mode, observed discharge is passed downstream at each sub-catchment outlet. If the observed
discharge is missing, post-processed streamflow is used instead of observed. In forecast mode, post-processed streamflow is
passed downstream. Note that ERRIS accounts only for uncertainty from the hydrological modelling component of the system,
and not uncertainties in rainfall forecasts. For the streamflow forecasts to be reliable overall, uncertainty from ERRIS must
sum to the uncertainty from rainfall forecasts.

### 2.2.3 Cross-validation and forecast verification

A leave-two-year-out cross-validation approach (Hapuarachchi et al., 2016) is implemented for all catchment models using
observed hourly data from 2007 to 2016. The first year of the leave-out period in each iteration is used for model validation.
The purpose of the second year is to avoid propagating any hydrological effects from the validation period into the model
calibration to make it independent (Hapuarachchi et al., 2016). A longer leave-out period is preferred, but this would shorten
the available data for model calibration. The duration of two years is determined as appropriate after considering the limited
data available. This approach is applied to all catchment models (Fig. 1a). Once a model is validated, we use the whole dataset
to calibrate the model to obtain the final parameter set for each sub-catchment. Raw and post-processed streamflow forecasts
are generated using the post-processed super-ensemble rainfall forecasts (Table 2) for the 36 months from 2014 to 2016
(Fig. 1b). Streamflow forecasts before and after error modelling are independently evaluated for different lead times at the
outlets of the selected 100 catchments for bias, accuracy, and reliability, as described in section 2.5.






**Table 2.** Forecast evaluation experiments

| Rainfall product | Ensemble members (raw) | Rainfall evaluation | | Streamflow evaluation | |
|---|---|---|---|---|---|
| | | Raw | Calibrated | Raw | ERRIS |
| ECMWF | 50 | Yes | Yes | | |
| ACCESS-GE2 | 24 | Yes | Yes | | |
| Super-ensemble | 75 | Yes | Yes | Yes | Yes |

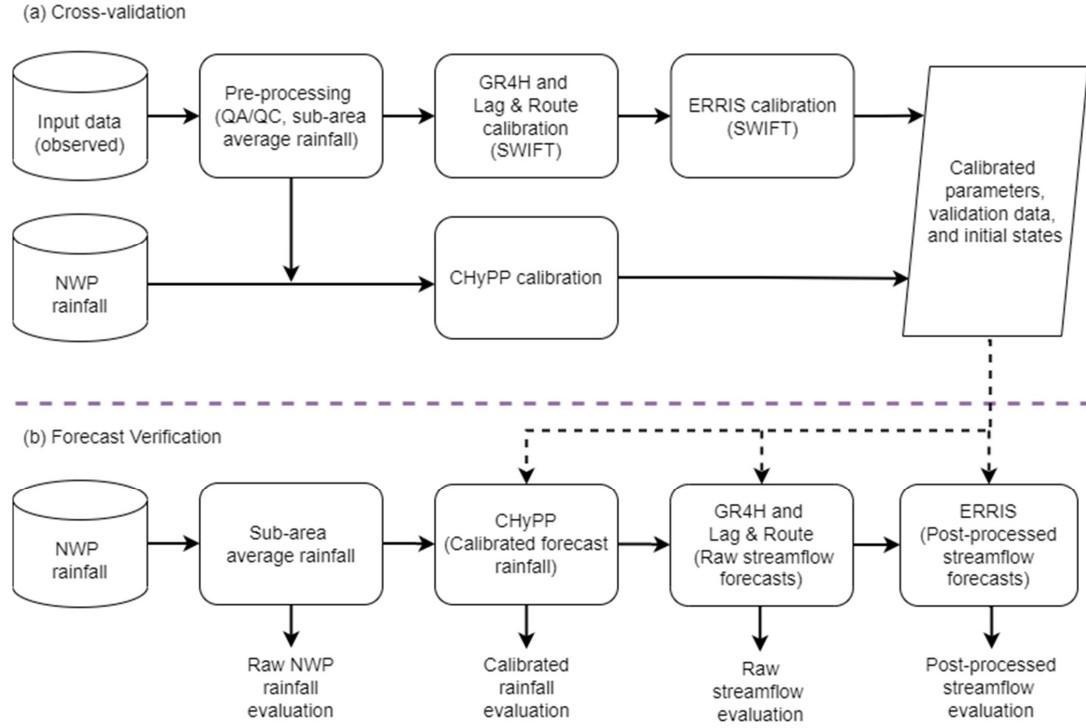


**Figure 1.** Rainfall and streamflow forecast evaluation framework: (a) Cross-validation, (b) Forecast verification

### 2.3 Determination of optimal ensemble size

It is essential to optimise computational efficiency and storage requirements of an operational system without compromising forecast quality. We conduct a sensitivity analysis using 6 catchments, located in different hydroclimatic regions, to estimate





the smallest ensemble size that does not significantly reduce critical measures of forecast performance. We set the maximum ensemble size to 1000 members based on the operational computational capacity. ACCESS-GE2 and ECMWF are calibrated to generate 1000 forecast rainfall members for each product using the hindcast data from 2014 to 2016 (1096 days). Then we generate a 1000-member streamflow forecast, corrected with ERRIS, from the rainfall hindcast. We randomly select $m$ (<1000) ensemble members from the 1000-member streamflow ensemble dataset (without removing ensemble members) and repeat

the process 100 times. In this exercise, $m$ is 50, 100, 200, 300, and 500. Then we calculate the Continuous Ranked Probability Score (CRPS, see section 2.5) for the randomly selected samples and compare them with the CRPS of the original 1000-member sample. The optimal ensemble size is decided considering the computational efficiency and the statistical characteristics of the randomly selected samples compared to the original 1000-member sample.

**2.4 Streamflow forecast quality assessment**

Skill score (see section 2.5.5) is a measure of expected forecast skill for a particular forecast location over a specified time period. The CRPS is the metric used in this study. Streamflow forecast skill is calculated using a bootstrapping technique (Efron and Tibshirani, 1994) to obtain fair and reliable skill statistics. The bootstrapping is implemented to provide an understanding of the possible range of skills that might be realised over a long period of record when only a short record of hindcasts is available. We also calculate the reliability for verifying forecast quality. A sensitivity analysis is conducted as

described below to i) select the optimum block size used in the bootstrapping method; and ii) check the effect of the number of bootstrapping iterations on forecast skill. The steps taken are:

1.   ACCESS-GE2 and ECMWF are calibrated to generate $x$ forecast rainfall ensemble members of each product using the hindcast data from 2014 to 2016 (1096 days). For each day, there is an hourly rainfall forecast to 7 days lead-time.

2.   Calibrated ACCESS-GE2, ECMWF, and raw PME forecast ensembles are combined to generate a *OES*-member super-ensemble.

3.   From the rainfall super-ensemble in (2) generate *OES* members of hourly streamflow forecasts. This dataset has the dimension of *OES* x 24$lt$ x $d$ data points where $lt$ = 7 is lead time (days) and $d$ =1096 (days).

4.   Since we calculate the forecast skill per lead day, the hourly streamflow data is aggregated to daily values and

Continuous Ranked Probability Score (CRPS) per day is calculated using the *OES* ensemble members to generate a matrix *MC* with the dimension of $lt$ x $d$ CRPS values.

5.   Data in *MC* per lead day is bootstrapped to calculate forecast skill. We randomly and iteratively select a block of data from the *MC* for each lead-day $p$ times such that the total data points are equal to $d$ and calculate the Continuous Ranked Probability Skill Score (CRPSS, see section 2.5). For an initial investigation, block sizes explored in this

study are a week and a month. From now on, we refer to the block sizes, w-block for a week and m-block for a month. For example, if the block size is a month, then $p$ is 36 (ie. 1096/ave no. days per month).

6.   Repeat step (5) for $k$ times where $k$ is 100, 200, 500 and 1000.





We do not select a block size of one day as the high autocorrelation of daily samples means they are not sufficiently independent.

**2.5 Verification metrics**

**2.5.1 Bias**

It is important to assess model bias to ensure the model is not consistently underestimating or overestimating streamflow. Bias (*Bias*) can be positive (underestimation) or negative (overestimation), and is calculated for each lead time using:

$$Bias\ (\%) = \frac{\sum_{i=1}^{n}(G_i - S_i)}{\sum_{i=1}^{n} G_i} \times 100 \tag{1}$$

where $G$ is observed value (rainfall or streamflow), $S$ is simulated/forecast (median of the ensemble) value, and $n$ is total number of observations.

**2.5.2 Nash-Sutcliffe Efficiency (NSE)**

The Nash-Sutcliffe efficiency (*NSE*) quantifies the relative magnitude of residual variance compared to the measured data variance, by:

$$NSE = 1 - \frac{\sum_{i=1}^{n}(G_i - S_i)^2}{\sum_{i=1}^{n}(G_i - \bar{G})^2} \tag{2}$$

where $\bar{G}$ is mean observed streamflow. In this study, NSE is used to assess the quality of GR4H streamflow simulations (deterministic), not forecasts.

**2.5.3 Continuous Ranked Probability Score (CRPS)**

CRPS measures the error of all ensemble members with respect to observations by integrating the squared distance between
forecast and observed cumulative distribution functions (Hersbach, 2000), and is given by:

$$CRPS = \frac{1}{T}\sum_{t=1}^{T}\int_{x=-\infty}^{x=\infty}(F_t^f(x) - F_t^o(x))^2 dx \tag{3}$$

$$Relative\ CRPS\ (\%) = RE = \frac{CRPS}{\bar{G}} \times 100 \tag{4}$$

where $F$ is the cumulative distribution function (CDF), $F_t^f(x)$ is the forecast probability CDF for the $t^{th}$ forecast case and $F_t^o(x)$ is the observed probability CDF (Heaviside function) and $T$ is the number of forecasts. Smaller CRPS values are the
better, and CRPS tends to increase with increased (positive or negative) forecast bias. For a deterministic forecast, the CRPS is replaced with the mean absolute error (MAE), which is the limiting value of CRPS when forecast spread tends to zero. The relative CRPS is represented as % of daily observations. Relative CRPS standardises errors to allow easy comparison between catchments.



### 2.5.4 Probability Integral Transform (PIT) uniform probability plots

We use the probability integral transform uniform probability (PIT) diagram to assess the reliability of ensemble forecasts (Laio and Tamea, 2007). PIT is uniformly distributed for reliable forecasts. It is the CDF of the forecasts $F_t(f_t)$ evaluated at observations $G_t$ and is given by:

$$PIT_t = F_t(G_t) \tag{5}$$

The empirical CDF of the PIT values falls on the 1:1 line when the forecasts are perfectly reliable. Deviation from the 1:1 line
indicates a less reliable forecast. To summarise and compare PIT values for many catchments, we use PIT-alpha (Renard et al., 2010) according to:

$$\alpha = 1 - \frac{2}{T}\sum_{t=1}^{T}\left|PIT_t^* - \frac{t}{T+1}\right| \tag{6}$$

where $PIT_t^*$ is the sorted $PIT_t$. An *alpha* value of 0 indicates the lowest reliability and 1 indicates perfect reliability. As the minimum rainfall amount measurable by tipping bucket rain gauges is 0.2 mm, we have set rainfall values less than 0.2 mm
as censored data for PIT calculation.

### 2.5.5 Continuous Ranked Probability Skill Score (CRPSS)

Skill is a measure of relative improvement of the forecast over a reference forecast. The Continuous Ranked Probability Skill Score is given by:

$$CRPSS = 1 - \frac{CRPS_{forecast}}{CRPS_{reference}} \tag{7}$$

where $CRPS_{reference}$ is the reference forecast. For this study, we use climatology as the reference forecast. Data from 1990 to 2016 are used for climatological streamflow calculation. For any given day of the year, the climatology value is the median of the period from 2 weeks before that day to 2 weeks after (i.e. 29 days) over the climatology period excluding the forecast year.

## 3 Catchment selection and data

### 3.1 Catchment selection

Australia has several climate zones as defined by Köppen Climate Classification (Stern et al., 2000; Peel et al., 2007), including equatorial, tropical and subtropical regions in the north and temperate regions in the south. The vast interior regions are covered by grassland and desert. There are 13 drainage divisions (Fig. 2), and mean annual rainfall for these divisions varies from 276 to 2816 mm (Table 3) calculated using data for the period from 1990 to 2016. Annual average potential evaporation (PE) is generally higher than annual average rainfall in most areas. Therefore, streamflow generation processes are mainly controlled
by water-limited environments (Milly et al., 2005) except for the Tasmania division. The pattern of rainfall-runoff and PE distribution within a year across different drainage divisions vary significantly. In the southern part of Australia, the wet season





begins in June-July and ends in December-January, while in the northern part of Australia, the wet season starts in November-December and ends in March-April (Bureau of Meteorology, 2021).

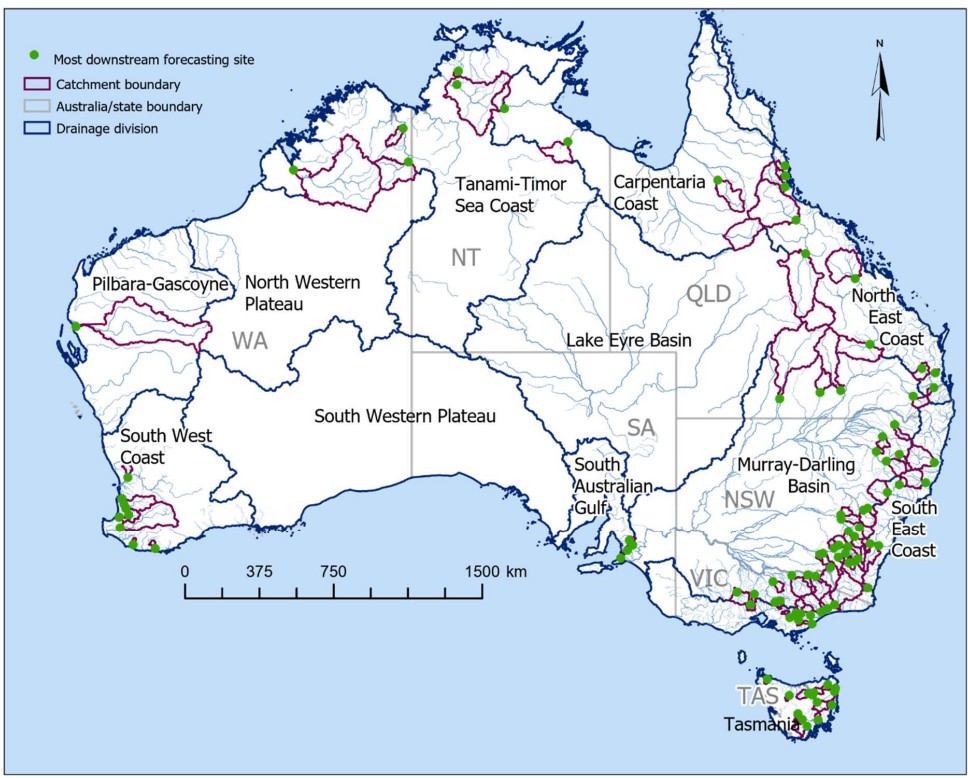


**Figure 2.** Map of Australia showing forecast locations, catchment boundaries and drainage divisions

For development of the SDF service, we select 100 catchments in consultation with the state and federal government entities and water management agencies in different jurisdictions and consider their strategic value (high economic, environmental, and social significance), data availability and other factors that support developing a successful model.

Most catchments are in the coastal regions (Fig. 2), covering most of Australia's populated centres. For the operational service, 281 potential forecast locations are identified within the selected catchments. There are no forecast locations selected in the Southwestern Plateau, Lake Eyre and North-western Plateau Divisions (Fig. 2), because there is no





significant user demand, and the gauging network is very sparse. Testing and verification of the modelling methodology
is done for the outlets of the selected 100 catchments. The same methodology is implemented for modelling all forecast
295 locations within a given catchment.

**Table 3.** Drainage divisions and catchment attributes

| Drainage division | No. of catchments | No. of forecast locations | Area (km²) | | Mean annual rainfall (mm) | | PE (mm) | | Aridity Index | |
|---|---|---|---|---|---|---|---|---|---|---|
| | | | Min | Max | Min | Max | Min | Max | Min | Max |
| North East Coast | 11 | 22 | 240 | 35985 | 549 | 2816 | 1504 | 1973 | 0.28 | 1.79 |
| South East Coast | 18 | 48 | 88 | 13700 | 605 | 1328 | 988 | 1431 | 0.46 | 1.34 |
| Tasmania | 14 | 28 | 227 | 3445 | 584 | 1530 | 758 | 978 | 0.65 | 2.02 |
| Murray-Darling Basin | 35 | 76 | 364 | 43720 | 462 | 1270 | 1048 | 1909 | 0.25 | 1.08 |
| South Australia Gulf | 3 | 4 | 345 | 701 | 570 | 825 | 1243 | 1293 | 0.44 | 0.66 |
| South West Coast | 9 | 21 | 26 | 18971 | 407 | 1034 | 1199 | 1625 | 0.29 | 0.70 |
| Pilbara-Gascoyne | 1 | 3 | - | 71222 | - | 276 | - | 2103 | - | 0.13 |
| Tanami-Timor Sea Coast | 5 | 19 | 658 | 83100 | 707 | 1567 | 2179 | 2259 | 0.32 | 0.72 |
| Carpentaria coast | 4 | 5 | 6020 | 17148 | 463 | 1062 | 2092 | 2252 | 0.22 | 0.47 |

**Note**: Statistics are calculated for the period from 1990 to 2016 using data from all forecast locations in the operational service. Catchment
mean annual rainfall is calculated using the hourly rainfall at sub-area centroids computed by interpolating the rainfall of the nearest four
gauges. Min and max rainfall and PE, and the Aridity Index, are calculated from the mean values of rainfall and PE of the catchments within
300 a drainage division. The Pilbara-Gascoyne Division has only one catchment, thus min=max. Three drainage divisions with no forecast
locations are not included in the table.

### 3.2 Observed data

This study collates relevant historical observations for consistent retrospective analyses across all catchments. Hourly observed
streamflow (1990 to 2016) and rainfall data (2007 to 2016) are extracted from the Bureau's internal databases and external
305 data provided by water agencies. Due to the limited availability of hourly rainfall data before 2007, the daily rainfall data are
extracted from the Australian Water Availability Project (AWAP, Raupach et al., 2009) and disaggregated to hourly time-
steps by linear interpolation. The disaggregated hourly rainfall data (1990 to 2006), is used for hydrological model warm-up,
since the quality of disaggregated rainfall data is low at the hourly scale. We have shown previously that disaggregated daily
rainfalls can provide good estimates of states in hourly hydrological models (Bennett et al. 2016). Rainfall and streamflow


observations are quality-controlled by comparing them to nearby stations and maximum station values, checking with data from different sources, and then removing any suspicious values using a semi-automated workflow. Streamflow is further checked for rating issues, the rate of change, and continuous zero values because in some ocations, missing values are replaced with zeros. The average areal observed rainfall for each sub-area is calculated using the inverse-distance squared-weighted averaging method, where the distance is calculated between the rainfall gauge and the sub-area centroid. Monthly gridded PE

data (1990 to 2016) at each sub-area centroid is extracted from the AWAP. PE is first disaggregated to daily values by assuming that monthly mean PE occurs in the middle day of each month, then linearly interpolating between these mid-monthly values. Note that this PE disaggregation method ignores the patterns of diurnal cycle and any correlation (negative) with rainfall. However, we note that the method is adequate for this study as Andréassian (2004) showed that GR4J is less sensitive to changes in PE inputs. To generate streamflow forecasts, we use climatological averages of PE calculated over the period 1990

to 2016.

## 4 Results

We cross-validate parameters for GR4H, channel routing, and ERRIS for each of the 100 catchment models. We found 97 of 100 forecast locations exceed the NSE value of 0.6 in the model validation. Catchments with NSE values lower than 0.6 contain intermittent or ephemeral rivers (Table 4). This may be partially due to the lack of representation of nonlinear dynamics

in the ephemeral catchment hydrologic processes, including the interaction between groundwater and stream channel, in the GR4H conceptual model. Below, we present detailed results of the experiments (Table 2) to identify the optimal ensemble size, effects of statistical processing, optimal parameters for the bootstrapping method, and acceptance criteria for selecting forecast locations for the operational service.

### 4.1 Optimal ensemble size

The number of ensemble members needed to maintain an acceptable forecast skill (see acceptance criteria in section 4.5) is important for an operational service as the service will generate a large volume of data through daily updates for 100 catchments. Optimal ensemble size is a balance between preserving the statistical properties while not creating unduly large data volumes. We implement the methodology described in section 2.3 to find the optimal ensemble size. The results are consistent across all the selected catchments, so for simplicity, we present results for one catchment. Figure 3 shows the

sensitivity of ensemble size on streamflow forecast accuracy (CRPS) with reference to a 1000-member sample for the Tully catchment (QLD). Streamflow forecasts using the ACCESS-GE2, and ECMWF products show similar accuracy for the same sample size (Fig. 3a and 3b). Generally, the forecast accuracy is proportional to the ensemble size. However, the relative increment of forecast accuracy is inversely proportional to the ensemble size. The overall results indicate that more than 200 members are sufficient to preserve greater than 98% statistical properties of simulated streamflow timeseries. Noting that

multi-model rainfall forecasts provide complementary benefits, and improve the streamflow forecast quality and the robustness





of the operational forecast system when the reporting of an NWP product is delayed or unavailable, we recommend using 200 calibrated members ($x$=200, section 2.4) of each rainfall forecast product (ACCESS-GE2 and ECMWF), for generating streamflow forecasts for the operational service. In the rest of the paper, we use calibrated ACCESS-GE2 and ECMWF, and raw PME merged product called super-ensemble ($OES$=401 members) for streamflow forecast evaluation. The streamflow
forecast skill of the super-ensemble is present in the section 4.3.

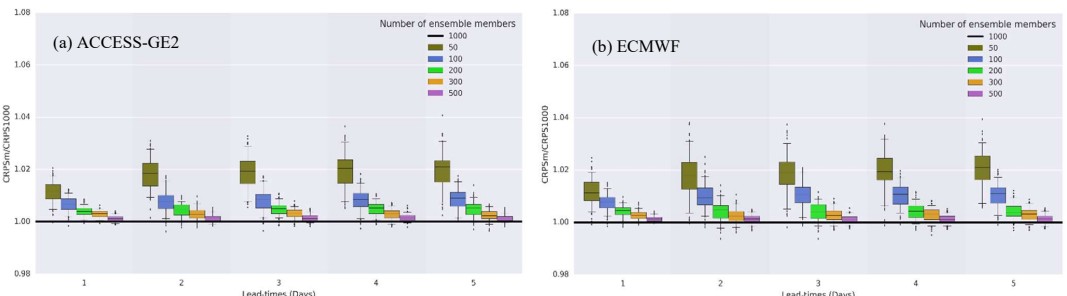

**Figure 3.** Sensitivity of ensemble size to forecast accuracy with reference to a 1000-member sample for rainfall forecasts (a) ACCESS-GE2 and (b) ECMWF for the Tully River at Euramo site (QLD).

**4.2 Effect of rainfall calibration**

Results for the rainfall evaluation across lead times, day-1 to day-7 (daily total), are presented using boxplot diagrams (Fig. 4-6). For each lead time, there are six boxes representing two raw rainfall products, ECMWF and ACCESS-GE2 and the super-ensemble, and their respective calibrated rainfall products. Figure 4 shows the bias (%) of different raw and calibrated rainfall forecast products for different lead times for the 100 catchments. Bias (%) is calculated for the ensemble median. Among the
raw rainfall forecast products, the ECMWF forecasts show smaller bias across most catchments. The bias of ACCESS-GE2 for different catchments is found to be more variable compared with ECMWF. For raw rainfall, bias increases with lead-time, whereas for calibrated rainfall, the bias variation with increasing lead time is marginal. Calibrated rainfall forecasts show a significant improvement of bias across all catchments and lead times, irrespective of rainfall product, location, and catchment size as indicated by the greatly reduced variation in bias. Also, the calibrated super-ensemble is less biased across the
catchments than the calibrated ACCESS-GE2 or ECMWF alone (Fig. 4). The bias correction using the CHyPP modelling approach is more sophisticated than only correcting mean bias of the rainfall ensemble. CHyPP utilises different marginal distributions, and *log-sinh* transformed bivariate normal distribution for the raw NWP rainfall forecasts and observed data, which allows for a non-linear bias correction (Robertson et al., 2013) resulting in much reduced variability of bias in the calibrated forecast rainfall.






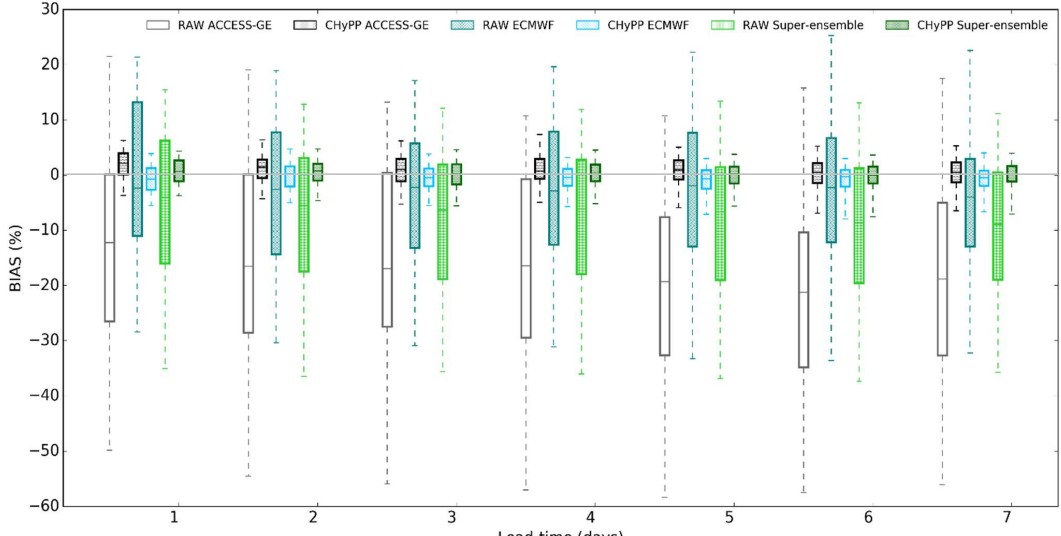

**Figure 4.** Bias (%) of raw and calibrated (using CHyPP) forecast rainfall products, ACCESS-GE2, ECMWF, and the super-ensemble for the 100 catchments.

Figure 5 shows reliability (PIT-alpha) of the different rainfall products. Among the raw rainfall forecast products, ascending order of reliability across a majority of catchments for all lead times is ACCESS-GE2, ECMWF, and the super-ensemble. Similar to bias (%), calibration substantially improves forecast reliability for all lead times for the tested rainfall products regardless of different catchment characteristics. Overall, calibration improves bias and reliability for all lead times.





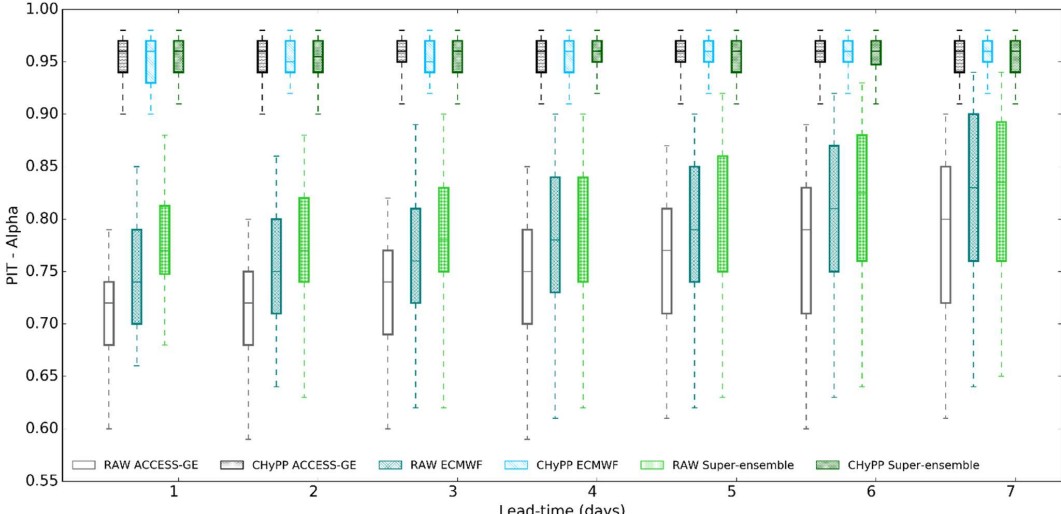

**Figure 5.** Reliability (PIT-alpha) of different rainfall products (raw and calibrated using CHyPP), ACCESS-GE2, ECMWF, and the super-ensemble for the 100 catchments.

The CRPS (error) value is highly related to catchment characteristics. For example, catchments having considerably long dry periods have numerically low average daily CRPS values. Therefore, we present the relative CRPS error (RE) where it is estimated by dividing CRPS by the mean rainfall and converting to a percentage value. Figure 6 shows the distribution of RE for raw and calibrated rainfall forecast products with lead-time, where lower error values are better. As expected, RE for all rainfall forecast products (raw and calibrated) increases with lead-time while the spread of the error distribution for raw rainfall reduces with lead-time. A narrower spread of error distribution over the forecast horizon is observed for all calibrated rainfall products compared to the raw data. Calibration reduces RE at shorter lead times but makes it slightly higher than raw rainfall at long lead times, while increasing the reliability significantly (Fig. 5). Normally calibrated rainfall values are closer to climatology values at long lead times, and there is a trade-off between sharpness and reliability.





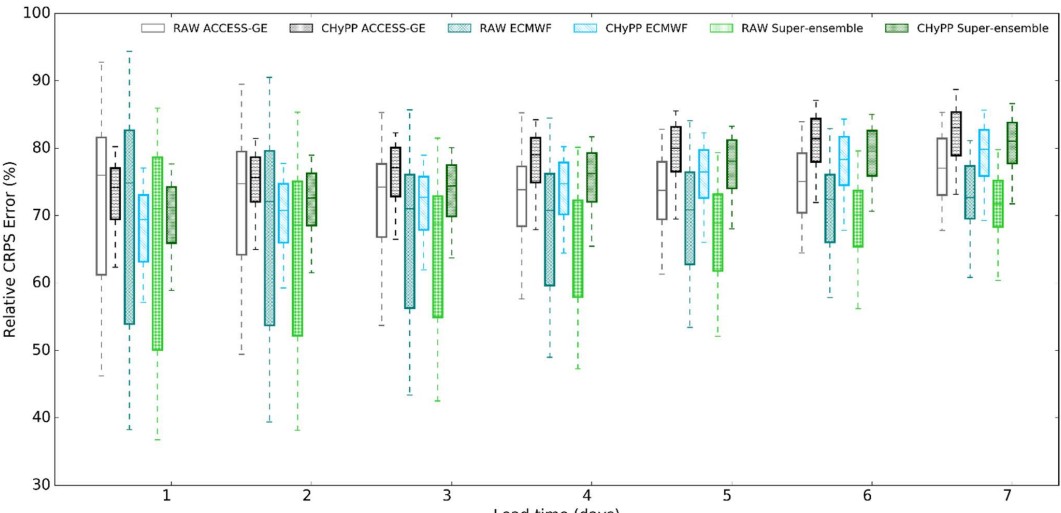

**Figure 6.** Relative CRPS (% of daily observations) of different rainfall products (raw and calibrated using CHyPP), ACCESS-
GE2, ECMWF, and the super-ensemble for the 100 catchments.

### 4.3 Effect of streamflow error modelling

Figure 7 shows bias (%) of streamflow generated before and after error modelling using the ERRIS model with the forecast
rainfall super-ensemble for 100 catchments. Bias (%) is calculated for the ensemble median. The bias increases with lead-time
for both raw and ERRIS-corrected streamflow forecasts. ERRIS-corrected streamflow forecasts (PSF) demonstrate relatively
low bias consistently across all the lead times compared to the raw streamflow forecasts (RSF) though the magnitude of
reduction varies across the continent. PSF show significantly reduced bias at short lead times for all forecast locations. For
lead day-1, median bias is less than 25% for all the forecast locations (Fig. 7a), whereas for lead day-7, the median bias is less
than 40% for about 40% of forecast locations (Fig. 7b).

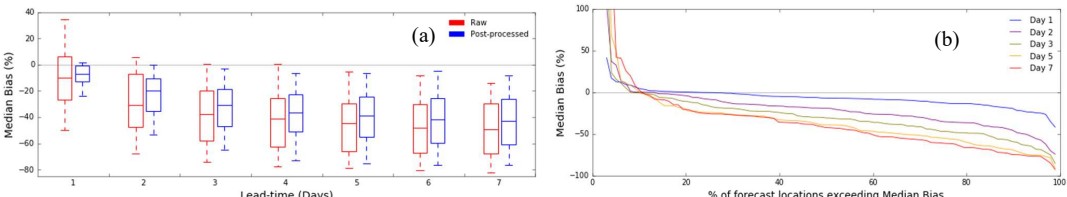





**Figure 7.** (a) Median bias (%) before (raw) and after streamflow error modelling with ERRIS for the 100 forecast locations; and (b) percentage of forecast locations not exceeding median bias (%) for ERRIS-corrected streamflow forecasts for different lead times.

The reliability of streamflow forecasts across all catchments is significantly improved consistently over the lead times with ERRIS (Fig. 8a), but the improvement is more prominent for the first three days. This improvement could partially be attributed to i) the effect of streamflow error modelling using ERRIS and ii) the improvement in the reliability and reduction of bias in rainfall forecasts. However, the reliability across different catchments, which are located in different hydroclimatic regions (Fig. 4), varies significantly; PIT-alpha is >75% for more than 80% of the catchments (Fig. 8b). A wide range of reliability across different forecast locations indicate that ERRIS performance highly relates to specific catchment hydro-climatic characteristics (Table 4).

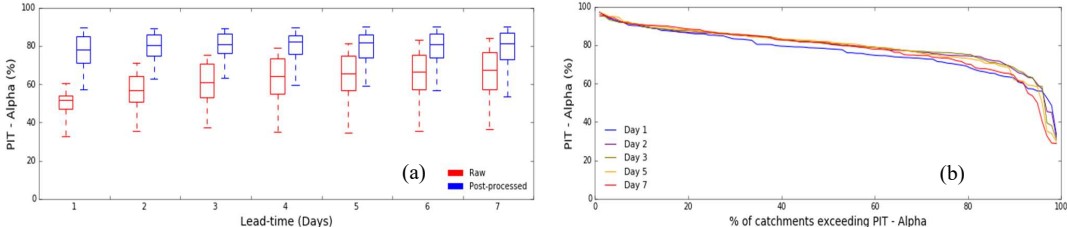

**Figure 8.** (a) Reliability (PIT) before and after streamflow applying ERRIS for the 100 forecast locations; and (b) percentage of forecast locations exceeding PIT (%) for ERRIS-corrected streamflow forecasts for different lead times.

Forecast skill (CRPSS) reduces with lead-time for both raw and error-modelled streamflow forecasts (Fig. 9). The degree of improvement of forecast skill provided by error modelling decreases with lead time (Fig. 9a). For lead day-1, all the forecast locations exceed 50% CRPSS (Fig. 9b) for error-modelled streamflow. Positive CRPSS means the forecast is considered better than using climatology. For lead day-7, CRPSS is positive for 60% of the forecast locations and it is 80% for lead day-6. Out of about 40% forecast locations where CRPSS is negative for lead day-7, most are located where observation networks are sparse and the rivers are intermittent or ephemeral due to dry climates. Further details are present in the discussion section.





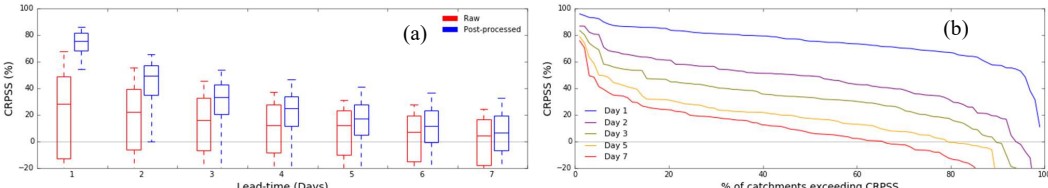

**Figure 9.** (a) CRPSS (%) of streamflow for raw and ERRIS-corrected forecasts; and (b) percentage of forecast locations (total
100) exceeding CRPSS (%) for ERRIS-corrected streamflow forecasts for different lead times.

### 4.4 Streamflow forecast skill

Forecast skills for hydrologic models are generally low for extreme events that rarely occur and only a few extreme events are
present in a relatively short period (e.g. 3 years) whin the datased used here. Bootstrapping allows exploration of model forecast
skill for a combination of various conditions, such as prolonged wet and/or dry periods, where the sequence of events (e.g.,
continuously a few wet or dry events) could be absent in the original dataset. Bootstrapped samples of the 3-year dataset might
contain more or fewer wet or dry periods than in the original 3-year dataset, thus providing a better indication of skill variability
across a more realistically varying sample. We test the methodology described in Section 2.4 for six catchments. Similar results
are found for all the catchments. For the explanation of results, Figure 10 shows bootstrapped forecast skill for the Acheron
River at Taggerty site for different number of iterations and block sizes. Forecast skill (CRPSS) is sensitive to block size (Fig.
10), and it reduces with lead-time for both the weekly-block (w-block) and the monthly-block (m-block) sample sizes. Forecast
skill calculated using w-block (Fig. 10a) shows less variation and narrower spread with lead-time than when m-block is used
(Fig. 10b). For the catchments we tested, the forecast skill is independent of the number of iterations for the w-block size
(Fig. 10a). For the m-block size (Fig. 10b), the forecast skill varies with the number of iterations. There is marginal variation
in the spread of the skill for iterations 500 and 1000 compared with iterations 100 and 200. This implies that the m-block size
captures uncertainty in the forecasts slightly better than the w-block size. Also, the m-block requires fewer computation
resources. Therefore, we adopt an m-block size for calculating the forecast skill for the operational service. There is no
significant variation of the results for a different number of iterations for both block sizes. This result may be partially attributed
to the small sample size of forecast data. To make sure we properly capture uncertainties in skill score calculation, we adopt
500 iterations for the operational service. Note that the block size may be catchment dependant, e.g., on catchment
450    characteristics such as geomorphology, hydro-climatology, upstream area etc. An alternative way of defining a block could be
by identifying wet, dry, and normal periods from the original dataset for bootstrapping. However, this process is unique to
each catchment and it is time consuming to implement for an operational service. Further research is required to investigate
block size dependency on catchment characteristics.

455





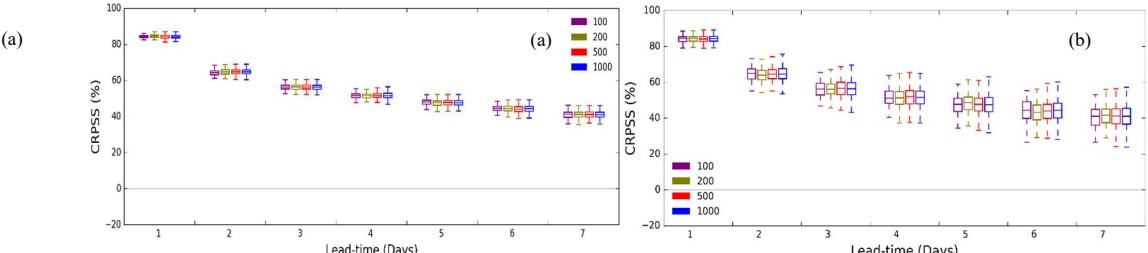

**Figure 10.** Bootstrapped forecast skill for the Acheron River at Taggerty site for different number of iterations and block sizes - (a) weekly; (b) monthly.

### 4.5 Acceptance criteria

It is essential for an operational service to maintain a certain standard for the quality of products provided to the users. In consultation with key stakeholders, we developed criteria, based on model performance and forecast skill, for accepting forecast locations for the operational service. The first criterion is that the Nash Sutcliffe Efficiency (NSE) of simulated streamflow is greater than 0.6 for the forecast location in the model validation (see section 2.2.3). This ensures the hydrological model is robust and produces acceptable results with observed data. If the first criterion is met, then forecast skill (CRPSS, see section 2.5), with reference to climatology, should be consecutively positive up to three days lead-time (Fig. 11). We calculate model performance metrics for each forecast location, and if the criteria for a forecast location are satisfied, it is added to the public service. If only the first criterion is satisfied, we consider releasing the forecasts only to registered users based on stakeholder requirements and the social and economic importance of forecasts at the location. If the first criterion is not met, then the forecast location is unsuitable for the service, and further revision of the model is required. We model 281 potential forecast locations in 100 catchments for the current service, and of these, 209 forecast locations in 99 catchments pass the acceptance criteria and are released to the public. On users' request, a further 17 forecast locations (including one additional catchment) with forecast skill slightly below the acceptance benchmark, are released to registered users only due to the economic and social significance of the forecasts.

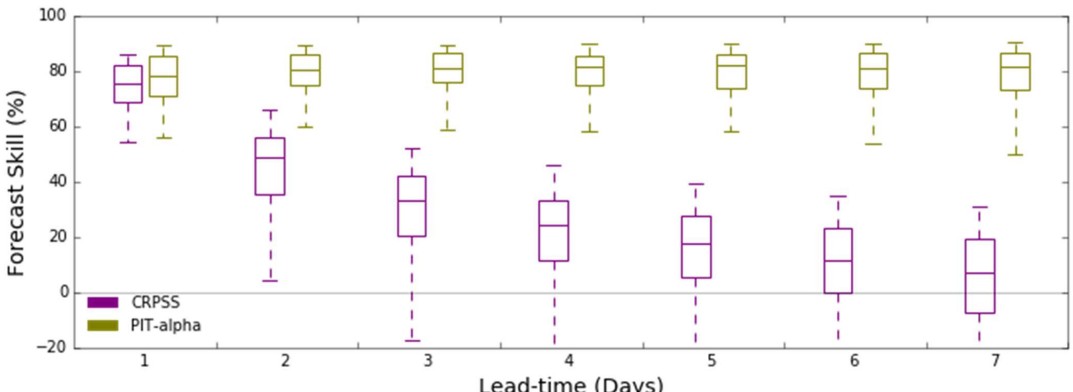

**Figure 11.** Forecast skill (CRPSS%) and reliability (PIT-alpha%) for the 209 forecast locations in the operational public service.

## 5 Discussion and future work

### 5.1 Interpretation of forecast skill

Model performance statistics of validation and forecast verification for lead day-3 for 281 potential forecast locations in the
seven jurisdictions of Australia are shown in Table 4. In model validation, SA models have the poorest NSE compared to other jurisdictions. Overall, 40% of forecast locations in South Australia and 23% in Western Australia fail the first acceptance criterion (NSE>0.6) while it is less than 12% for other jurisdictions. We note that some forecast locations in WA, TAS, and inland areas of NSW, QLD, and VIC show poor NSE. These areas have intermittently flowing rivers due to arid or semi-arid climates (see the Aridity Index in Table 3). Much of continental Australia to the west of the Great Dividing Range (an area of
>5 million km$^2$) is sparsely populated and characterised by intermittent and ephemeral streamflows. Therefore, the observation network is also sparse and there is not enough benefit to justify the cost for expanding the observation network. Ephemeral rivers are subject to strongly non-linear relationships that are less well understood in rainfall and runoff and are inherently more challenging to model than perennial catchments (Gutierrez-Jurado et al., 2021). The forecast skill (CRPSS) is also poor for SA and inland parts of TAS, NSW, QLD and VIC. This is partially due to the poor quality of rainfall forecasts (Shresta et
al., 2013). Arid regions are generally characterised by high rainfall variability, and often these rainfalls are underestimated by NWP models. It may be difficult for NWP models to replicate the complex meteorological processes that drive the high rainfall





variability with limited observations. Therefore, improving forecast skill in ephemeral catchments is likely to remain challenging.

**Table 4.** Hydrologic model performance statistics for 281 forecast locations

| Jurisdiction | No. of forecast locations | Validation | | | | Forecast (for lead day-3) | | | | | | | |
|---|---|---|---|---|---|---|---|---|---|---|---|---|---|
| | | NSE (%) | | | | CRPSS (%) | | | | PIT-alpha (%) | | | |
| | | 5% | Median | 95% | Max | 5% | Median | 95% | Max | 5% | Median | 95% | Max |
| NSW | 69 | 48 | 80 | 95 | 98 | -22 | 29 | 53 | 61 | 59 | 80 | 89 | 91 |
| NT | 17 | 65 | 87 | 97 | 99 | 25 | 43 | 68 | 69 | 64 | 79 | 87 | 87 |
| QLD | 42 | 56 | 82 | 97 | 98 | -763 | 18 | 45 | 60 | 46 | 77 | 89 | 89 |
| SA | 9 | -56 | 66 | 81 | 84 | -344 | 3 | 10 | 12 | 9 | 79 | 93 | 93 |
| TAS | 33 | 41 | 79 | 94 | 97 | 3 | 24 | 41 | 44 | 59 | 83 | 94 | 95 |
| VIC | 72 | 48 | 77 | 95 | 98 | 2 | 37 | 62 | 77 | 65 | 79 | 92 | 95 |
| WA | 39 | 19 | 83 | 97 | 98 | 10 | 41 | 78 | 92 | 30 | 77 | 89 | 93 |

**Note:** y% is yth percentile, NSW: New South Wales, NT: Nothern Teritory, QLD: Queensland, SA: South Australia, TAS: Tasmania, VIC: Victoria, and WA: Western Australia. CRPSS and PIT-alpha values are the median of the respective bootstrapped samples.

NWP rainfall calibration using CHyPP reduces bias and increases reliability across all catchments (Fig. 4, Fig. 5). In doing so, there is a compromise in relative CRPS – an improvement in shorter lead times but there is no discernible improvement – or in some cases a slight decline - at longer lead times (Fig. 6). However, relative improvements in forecast skill in ephemeral catchments are less prominent compared with perennial catchments. Similar results were found by Li et al. (2021). These results are discussed with many stakeholders across the country as part of development of the operational service. A clear message from them is that reliable streamflow forecasts are more important than precise forecasts for long lead times to downstream users, and will be beneficial to their decision making.

Streamflow forecast skill calculated using the bootstrapping technique appears to be realistic for most forecast locations. This gives some confidence that we can expect similar performance under operational conditions. However, in this study, bootstrapping is only able to sample within the evaluation period, which is three years from 2014 to 2016. The years 2014 and 2015 were average to dry years for most of the selected catchments, which are located along Australia's coastal regions (Fig. 2). The year 2016 was a wet year for South Australia, Victoria and Tasmania, where about a half of the selected catchments are located. Overall, there were only a few wet events in the evaluation dataset. Therefore, we recommend that users are cautious when interpreting the forecast skill for wet events. A more extended period of data, with balanced wet and dry events, is recommended for a better evaluation of streamflow forecast skill. In addition, short-term verification statistics on daily basis will be useful for the users for better decision making.


We demonstrate with various performance measures that calibration adds value to raw NWP rainfall forecasts, and the relative improvement is different for each product. For example, raw ECMWF rainfall is less biased compared to ACCESS-GE2

(Fig. 4). Therefore, the selection of rainfall forecast products for the operational forecasting system may effect the quality of streamflow forecasts (see Section 6). In this study, our criteria for selecting NWP rainfall forecast products are availability of the product at the BoM, hindcast period, and ease of use (i.e. format, extent, file size). Where a range of suitable products are available, we recommend conducting a thorough evaluation before selecting NWP products to use.

### 5.2 Uncertainties in forecasts

We try to minimise input data uncertainty by calibrating NWP rainfall forecasts using the CHyPP model and minimise the hydrologic uncertainty by applying the ERRIS error model to simulated discharge. We demonstrate that calibrated NWP rainfall forecasts improve streamflow forecast skill. Similar results are found in Canada and South America (Jha et al., 2018; Rogelis and Werner, 2018). However, uncertainties may also arise from the observed data used to calibrate parameters in the hydrologic models. The most common issues are precision of the instruments that measure the water level (stage) and rainfall,

derivation of the stage-discharge relationship (rating tables), the accuracy of gauged rainfall interpolation methods (e.g. inverse-distance squared-weighted averaging), and data disaggregation methods. Measurement and rating curve uncertainties in streamflow, particularly for low and high flows, bring additional complexities in model calibration/validation and ultimately model performance (Tomkins, 2014). There are advanced methods for estimating the uncertainty caused by instrument and rating errors. Maldonado et al. (2018) present a method for estimating the uncertainty associated with stage-discharge relations

using Bayesian inference with the likelihood estimator approach. However, maintaining the precision of observation instruments and updating rating tables is the responsibility of data providers, including state water management entities, irrigators, hydropower generators and water utilities and is out of scope of this study. Although there are many complex methods available for climate data disaggregation (Breinl and Di Baldassarre, 2019; Görner et al., 2021; Mehrotra and Singh, 1998), for simplicity, we use a simple method, linear interpolation for disaggregating daily rainfall and PE data to hourly. PE

varies with the diurnal cycle and usually shows some degree of (negative) correlation with rainfall that could have been considered in the disaggregation. However, we note that the impact of rainfall uncertainty has been shown to be more significant than PE in hydrological modelling (Anon, 2011; Guo et al., 2017; Paturel et al., 1995). Therefore, the accuracy of spatially averaged gauged rainfall data through interpolation and disaggregation may have a greater effect on hydrologic model calibration. Perry and Hollis (2005) and Legg (2015) found the accuracy of gridded rainfall data depends on density of the rain

gauge network, with more significant errors associated with sparse gauge coverage. The sparseness of the rainfall observation network in much of inland Australia (particularly in the inland desert regions and in northwest Australia) remains a challenge for the development of any streamflow forecasting system.



### 5.3 Streamflow error modelling method

Modelling hydrological errors using the ERRIS model significantly reduces the bias and improves the forecast skill (Fig. 7).
Improvements in forecast skill depend on location, season, and lead time (Hegdahl et al., 2021; Jha et al., 2018). However, calibration of ERRIS is sensitive to the quality of observations (Li et al., 2016). ERRIS uses a *log-sinh* transformation to normalise streamflow prediction errors, and the transformation amplifies errors related to low simulated flow and modulating errors related to high simulated flow. Therefore, if there are large uncertainties in low streamflow observations, these will result in large residual variances in the transformed space and lead to large forecast uncertainties. As a result, forecasts may
be reliable, but have low precision particularly, at long lead times.

The ERRIS model applies corrections to hydrological model output, but it does not address the underlying cause of the forecast errors. Relatively simple error models like ERRIS try to characterise prediction errors that arise from many different causes and persist over many different time horizons. For example, error models may try to address: (i) long-term or average forecast errors related to the hydrological model calibration, (ii) forecast errors that persist for intermediate time periods of days to
months that may arise from the effects of errors in magnitude of catchment rainfall estimates for a significant event, and (iii) transient errors related to incorrectly assumed diurnal pattern in potential evapotranspiration or small errors in the timing of catchment rainfall. On the other hand, data assimilation methods seek to address underlying causes of some hydrological simulation errors, particularly those that persist over long and intermediate timeframes, by updating model state variables (initial conditions) and forcing, so that model predictions better reflect observations. However, implementation of a data
assimilation method for probabilistic streamflow forecasting in a semi-distributed modelling setup is challenging due to the complexity in inter-dependencies of uncertainty contributing sources such as an ensemble of model forcing data, model state variables and/or model parameters (Moradkhani et al., 2005; Li et al., 2016). In a forecasting context, the objective is to ensure that the initial condition set in a hydrological model better reflects reality and therefore forecast errors are likely to be smaller. However, even after updating state variables, hydrological model predictions are unlikely to be perfect, and therefore, a role
for an error model such as ERRIS in an operational system is still worth exploring. Decreased dependence on error corrections through weaker bias corrections, lower autocorrelation parameters, and lower residual variances when ERRIS is calibrated using updated streamflow forecasts by a data assimilation technique, may improve the overall streamflow forecast skill. Further exploration for implementing data assimilation for the service is planned.

### 5.4 Challenges in operational forecasting and opportunities

Over the last two decades, the number of studies in ensemble streamflow forecasting has increased significantly. However, applications of ensemble forecasting vary significantly in terms of geographical distribution, forecast horizon, methodology and evaluation. This could partially be due to the evolution of ensemble streamflow forecasting science from research to operations. There are many challenges in the large-scale operational adoption of ensemble streamflow forecasting (Pagano et



al., 2014; Wu et al., 2020). Some of the critical areas for future focus and consideration for ensemble streamflow forecasting research, operational application, adoption and benefit to the community are:

- **Best use of the available data:** In Australia, observed rainfall at a sub-daily time scale is available for most stations. However, the number of rainfall stations is declining over time, and some of the catchments already have a sparse network. Measurement of PE data is rare across the country. Simulated monthly PE data from the AWAP (Raupach et al., 2009) is disaggregated to hourly for hydrological model application. Streamflow gauging stations where the automated facility is available for reporting in real-time are ingested in to the BoM system. These observed data are the backbone of the hydrological model construction, calibration, validation, and forecasting. Any improvements in measurement and rating curve uncertainties may result in better performance in streamflow forecasting. Updating measurement stations with automation facilities may result in better quality data which could be useful for more skilful forecasting. This study demonstrates that improvements in NWP rainfall forecasts directly contributes to improvements in streamflow. Any further improvements in NWP rainfall forecasts will result in more accurate and reliable streamflow predictions. Possible improvements in different flow regimes, particularly low and high flows, will be explored in the future. Endeavours should also be undertaken to explore emerging science, including merging radar rainfall with NWP forecasts (Velasco-Forero et al., 2021).

- **Extending the forecast horizon:** In addition to the 7-day ahead forecasts, the Bureau also provides operational seasonal predictions (http://www.bom.gov.au/water/ssf/index.shtml) from one to three months ahead (Woldemeskel et al., 2018). Potentially the gap between these two forecast ranges could be minimised by extending the 7-day streamflow forecasts to multi-week forecasts. Rainfall forecast data to at least 30 days ahead are now available, and the multi-model ensemble approach could be used to increase the predictability and reliability of these rainfall forecasts (Specq et al., 2020). The potential use of the rainfall forecast data for extending the streamflow forecasts to 30 days ahead could be explored in the future. A novel Multi-Temporal Hydrological Residual Error (MuTHRE) model (McInerney et al., 2020) has been recently developed to enable reliable streamflow streamflow forecasting beyond one week. The model has been tested for 11 catchments to generate sub-seasonal forecasts (lead-time 1-30 days) using the GR4J hydrologic model and calibrated rainfall forecasts from the ACCESS-Seasonal NWP model (McInerney et al., 2020). They found that forecast performance was improved compared to the current seasonal streamflow forecasts in terms of sharpness, volumetric bias, and skill. This approach could be further explored for wider-scale applications for seamless streamflow forecasting.

- **Forecasting in managed river systems:** At present, the Bureau's operational streamflow forecasting services do not receive real-time and future water releases from dams and reservoirs. Therefore, the 7-day streamflow forecasting service is developed for catchments with minimal or no anthropogenic influences (e.g., releases from storage, extractions). Catchments in this study (Fig. 2) are all upstream of dams, reservoirs, or weirs, and have no significant water extraction or irrigation return flows. Further investigation to account for these anthropogenic processes will lead to greater expansion and application of the forecast service. Research should be conducted to understand how



these anthropogenic influences impact the forecasts and incorporate practical and innovative solutions into the hydrological forecasting models.

• **Effective communication:** The 7-day ensemble streamflow forecasting service produces large volumes of information. Therefore, key messages must be conveyed clearly and efficiently for correct interpretation, allowing for well-informed decision making and common understanding among end-user communities. The user communities may range from experts in water management in decision-making entities to those with no experience in using ensemble forecast products. To effectively communicate forecasts with end-users in mind, the BoM consults widely

and frequently with stakeholders, considers their needs, and provides clear and effective forecast visualisations, including the website and forecast products. The BoM continually improves the forecast products through stakeholder consultation and feedback.

• **Maintaining operational service:** Maintaining an operational service is a big task – and requires a well-trained, dedicated team of staff with expert knowledge of the catchments and experience with hydrologic model application

and forecast system configuration. Losing in-house modelling or systems expertise due to limited funding or incentives may result in suboptimal forecast quality and end-user benefits.

Each of these challenges shares a real-world perspective and opens ongoing research and development opportunities, resulting in a greater update of ensemble streamflow forecasting for operational decision-making.

**6 Description of the operational forecast system**

We develop the operational 7-day ensemble streamflow forecasting system based on the evidence derived form the above results. We design the SDF forecast system to use multi-model rainfall forecasts to improve the quality of streamflow forecasts and minimise the potential risk of system failure due to the absence of NWP rainfall input. The rainfall forecasts used in the SDF service are ECMWF and PME (Fig. 12). We also planned to use the ACCESS-GE2 product for the service and conducted

an extensive evaluation as presented in this paper. However, the operational delivery of the ACCESS-GE2 had been delayed, and therefore, it is to be included in the service later. The CHyPP model is used to calibrate ECMWF forecasts and generate 400 bias-corrected and statistically reliable hourly rainfall forecast members. We combine calibrated ECMWF and PME rainfall forecasts and input them into the SWIFT model to generate 401 members of hourly streamflow forecasts (Fig. 12). Ensemble streamflow forecasts are fed into a product generator to produce plots, tables, and data files and publish in a web

portal (www.bom.gov.au/water/7daystreamflow). In addition to the web plots, users can extract data through the web portal and ingest forecast data to their operational systems via a File Transfer Protocol (FTP) link. The forecasts are generated daily in the Bureau of Meteorology's operational platform, Hydrological Forecasting System (HyFS) (Robinson et al., 2016). It is the central national platform that supports flood forecasting and warning as well in Australia. HyFS is a Delft-FEWS (Flood Early Warning System) based forecasting environment (see http://oss.deltares.nl/web/delft-fews/about). HyFS allows



ingestion and processing of real-time observations and numerical weather prediction (NWP) model rainfall forecasts, running
routine workflows, model internal state management and forecast visualisation. The process is fully automated (Fig. 12), and
forecasts are updated daily between 10:00 AM and 12:00 AM AEST.

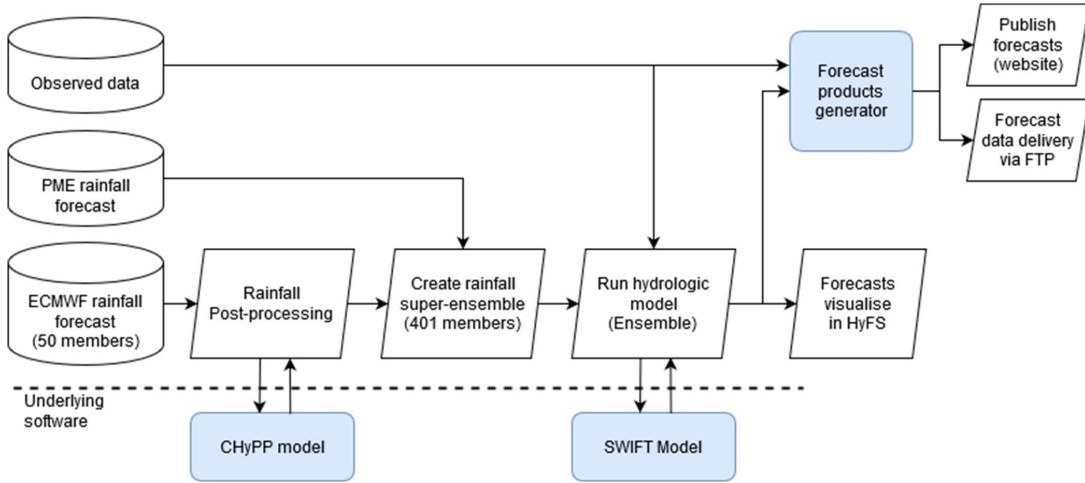

**Figure 12.** Operational forecast system (HyFS workflow level)


## 7 Summary and conclusions

We present the development of a 7-day ensemble streamflow forecasting service for Australia
(http://www.bom.gov.au/water/7daystreamflow/). The service has been operational since December 2019 and provides daily
updates of streamflow forecasts up to 7 lead days for 209 forecast locations in 99 catchments for the public and an aditonal 17
forecast locations including one catchment to the registered users. The forecast system is capable of ingesting and calibrating
multi-model ensemble NWP rainfall forecasts using the CHyPP model, which combines a Bayesian joint probability model
and the Schaake Shuffle method. Calibrated ensemble rainfall forecasts are fed into a hydrological modelling package, SWIFT,
which then generates error-corrected ensemble streamflow forecasts.

We show that calibrating NWP rainfall forecasts using the CHyPP model significantly reduces bias and improves relaibility.
Error modelling of streamflow forecasts using ERRIS further improves their accuracy and reliability. A sensitivity analysis
for optimising the number of streamflow ensemble members for the operational service shows that more than 200 members
are needed to represent the forecast uncertainty. We show that the bootstrapping block size is sensitive to the forecast skill
calculation and a month is better than a week as the monthly block size allows to capture maximum possible uncertainty.



Acceptance criteria is defined based on model validation and verification results for selecting locations with an adequate
forecast quality for the operational service. The acceptance criteria is defined as an NSE greater than 0.6 in model validation,
and a median of bootstrapped model verification skill (CRPSS) that is positive (greater than zero) for consecutive 3-days lead-
time. Incorporation of ACCESS-GE3 rainfall forecasts into the operational service is planned, and continued stakeholder
feedback will be used to guide further enhancements of the service.

## Acknowledgements

We acknowledge funding from the Water Information Research and Development Alliance (WIRADA) for SWIFT model
development. We thank colleagues in the Bureau of Meteorology from Water Forecasting Services, HyFS, systems support,
Data and Digital, and Science and Innovation for their support in developing this operational service. We acknowledge the
national and state water agencies across Australia for providing gauged data, critical catchment information for modelling, and
their input for forecast location selection and products design. The operational website was developed in consultation with the
CSIRO and approximately 70 other stakeholders across Australia. We would like to express our sincere thanks to our technical
reviewers Dr Beth Ebert and Dr Christoph Rudiger, and Dr Elisabetta Carrara for their time, careful review and valuable
comments and suggestions on the submitted version of this manuscript. The large computations in this study were conducted
using the facilities provided by the National Computational Infrastructure (NCI) supported by the Australian Government.

## Author Contributions (in alphabetical order)

HAPH designed all experiments with support from all authors. AK, MH, NG, FW, SZ, and HAPH contributed to data
extraction, quality checking, catchment delineation, model setup, and cross-validation. MH conducted all NWP rainfall
evaluation experiments and prepared results. AK conducted all streamflow evaluation experiments and prepared results. NG
conducted geo-spatial analysis and prepared maps. PS developed all workflows and modelling tools, forecast products
generator, web interface, modification to the HyFS, and data achieving system. HAPH and MAB analysed the results and
prepared the manuscript with contributions from all authors. PMF and MAB provided project administration, resource
allocation and scientific editing support. DR and JB contributed to the development of the methodology and reviewed the
paper.



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
