# Peer review of "Development of a national 7-day ensemble streamflow forecasting service for Australia"

_Hydrology and Earth System Sciences, 2022_

## Author Comment (AC1)

Hydrol. Earth Syst. Sci. Discuss., referee comment RC1
https://doi.org/10.5194/hess-2022-72-RC1, 2022

[Figure]

**Comment on hess-2022-72**

Anonymous Referee #1
* * *
Referee comment on "Development of a national 7-day ensemble streamflow forecasting service for Australia" by Hapu Hapuarachchi et al., Hydrol. Earth Syst. Sci. Discuss., https://doi.org/10.5194/hess-2022-72-RC1, 2022
* * *
**Overview**

This is a detailed descriptive article on the methodology followed to set up an Australian ensemble streamflow service. I commend the authors on the clear description and succinct summary of what I imagine was a very large project. I believe the submission would be of interest to readers of HESS, particularly due to the value of sharing the development of operational systems with the academic community.

*Author response: We thank the reviewer for commendation and acknowledgement.*

The paper is understandable heavily focused on Australia. I have a couple of suggestions which would help make this work relevant to a wider audience. Firstly, I suggest that more context is given to help the reader understand the hydro-climatic context that the model is being validated over for example by including some maps instead of / alongside the box blots and table summaries (further comments on this are detailed below). Secondly, I would like to see more discussion of how the development of this service in Australia builds on, and moves forward, the development of ensemble streamflow services around the world. At present the work is situated in the Australia context and the reader is given limited insight into what is novel or new about this work or why a particular approach is suitable for Australia but may not have been used elsewhere. A wider review of existing literature would help support this.

*Author Response: We thank the reviewer for their suggestions. We are happy to accommodate the reviewer's suggestions:*

- *We will elaborate the introduction section of the manuscript to give greater context to different hydroclimatic regions of Australia (lines 66-85) as suggested below, and introduce a new figure in Section 3.1 and elaborate it. We will also review existing literature and include this in the introduction and discussion sections.*
- *We will elaborate the discussion section and will give greater insight to what is unique and novel about this research and why it is suitable for Australia.*

From a technical perspective the work appears sound, an assessment of the strengths and limitations on the underlying data is made and a series of established verification metrics applied. The methodological steps are clearly documented throughout.

*Autor Response: Thanks for the acknowledgement, we highly appreciate it.*

From an open data perspective there is no indication of the source of quality of the observed rainfall and flow data.

*Author response: We collected rainfall and discharge data from the Australian Bureau of Meteorology databases. This data is partially quality checked.*
*The collected data goes through a comprehensive quality checking using a semi-automated workflow by visualising streamflow and nearby rainfall station data side by side. This allows the modeller to identify the connection between rainfall and streamflow (i.e. there should be a high rainfall event for high discharge). This approach assists the modeller to confidently make necessary corrections to the observed data. Then the quality checked data are visually checked (plots) for further quality assurance. The corrections/modifications made to the original data are recorded (a data file) so that all the users of this dataset are aware of them. If necessary, we can elaborate on this process in Section 3.*

My main technical concerns come from the representation of extremes within the skill assessment. L66-85 sets the context of hydrological extremes in Australia and identifies both floods and droughts as particular water management challenges. The representation of high and low flows in forecast systems leads to different challenges at different parts of the flow regime yet the discussion around model assessment does not address this as you use evaluation metrics across the full flow regime, it is well documented that it is much easier to model non-extreme flows. Is there also a need to consider the skill of the forecast system in identifying events that cross a high / low threshold as it is during these events that the system will have more operational value and your results may be skewed depending on characteristic of individual catchments. I appreciate the system is already operational and it may not be appropriate to add this to this paper, but it would be helpful to acknowledge this limitation and maybe identify it as a future research area.

*Author Response: We very much appreciate the reviewer's concerns regarding extreme events, and its performance evaluation using metrics presented in Section 2.5. We certainly agree to acknowledge this limitation and identify it as a future research area.*
*The Bureau provides more specific and specialised services for flood and drought forecasts. Therefore, if a flood is current for a location, the users of this service are redirected to Bureau's*

*operational flood forecasting and warning website (please see a screenshot shown below – yellow banner). Therefore, we deliberately avoided discussing extreme events in this paper. However, we appreciate reviewer's comments and agree to acknowledge this limitation and identify it as a future research area.*

[Figure]

**Specific comments on the text and figures**

L56 – 65 – it is unclear to me what this paragraph on continental and global scale models adds to the paper. Could you integrate this in the context of developing a streamflow model for Australia e.g. what lessons did you learn from the existing global models?

*Author Response: Thanks for the suggestions. We will elaborate the particular relevance to Australia.*

L98 – do you know of other examples of "hybrid dynamical-statistical streamflow forecasting systems" or similar set ups. It would be helpful here to identify if there is anything unique about the Australian system compared to other operational systems in other countries.

*Author Response: There are not many operational systems that we are aware of. The HEFS system (Demargne et al. 2014 - https://dx.doi.org/10.1175/bams-d-12-00081.1) is a similar one, but it applies a calibration to rainfall, and an error model. But in practice it uses 'in-the-loop' flood forecasters to do data assimilation manually (or at least, that was the case last we checked), which we imagine would mean it would struggle to produce reliable ensembles in operational setting. The EFAS (EFAS Post-processing - Copernicus Services - ECMWF Confluence Wiki) system uses dynamical models only, as does GloFAS system.*
*We will elaborate this and present the uniqueness of the system in the Australian context.*

Table 3 – for those not familiar with Australian climatology it would be helpful to show some of the info in this table graphically e.g. could you include a map of mean annual rainfall distribution (or anther representative variable) across Australia, it's hard to fathom this from the table, especially as the number of catchments in each drainage division are quite varied. Other information that might be interesting is an indication of the catchment response time, are you looking at steep flashy catchments or slowly responding catchments. Later on you

mention ephemeral rivers as a reason for lower forecast skill, again is there a particular region where they are more common? This type of characteristics information would help readers compare your approach to approaches taken in other countries and understand potential spatial variations in your model skill.

*Author Response: We acknowledge the reviewer's suggestions. We will introduce a new spatial map depicting annual rainfall distribution and forecast skill of the forecast locations. We will also discuss any connection between forecast skill and catchment characteristics as the reviewer has suggested.*

Fig 7 – the caption and x axis label for fig 7b are inconsistent

*Author Response: We acknowledge the error and will fix it.*

Section 4.5 Acceptance Criteria - How did you specify the 0.6 NSE threshold? Was this in conjunction with user requirements or based on existing published thresholds? Do you have any indication of the acceptable forecast skill for users? I find it interesting that there were additional sites when the forecast skill wasn't 'scientifically acceptable' yet users still wanted to receive this information. How have you addressed presenting forecast skill in the user interface? Also see my comments above re: the skill for different parts of the flow regime, did you incorporate this in any way?

*Author Response: We adopted NSE of 0.6 from Chew and McMahon (1993) in consultation with the stakeholders. We will elaborate this section and will include reference. We agree that sites where forecasts are not considered ''scientifically acceptable'' using this metric may still bring benefit to the user communities. The forecast skill criteria are one set of measures for selecting a forecasting location for the service. We consulted our stakeholders and identified forecast locations critical for their decision making and added to the service. Sometimes, the forecast locations with poor skill are only available to registered users. This is to reduce possible miscommunication with the public and to keep the reputation of the service. To address reviewer's question, we will elaborate Sections 4.5 and 5.4. Please find below a sample plot of forecast skill for a forecast location (with good skill) that is available for users via the service website. A description of how to read this plot is also given on the website. (http://www.bom.gov.au/water/7daystreamflow/#panel=advanced).*

[Figure]

Section 5.1 goes on to discuss some reasons for variability in forecast skill, could you show the forecast skill spatially on a map and any links to catchment/meteorological forecast characteristics? Again the table display in Table 4 is difficult to interpret due to the number of forecast locations lumped into each jurisdiction.

*Author Response: We acknowledge the reviewer's suggestion, and we will include another figure (map) showing forecast skills for different locations across Australia.*

Section 5 is interesting and raises established challenges of operational streamflow forecasting however it lacks integration with the rest of the paper. Possibly this could be improved with incorporation of wider literature on development of streamflow forecasting systems mentioned above. I also suggest it is moved after section 6 so that it links to the summary and conclusions section.

*Author Response: We acknowledge the reviewer's suggestions. We will elaborate Section 5 to integrate more closely with the results and Australian relevance as presented in previous sections. We will present Section 5 after Section 6.*

**Reference**

Chiew, F.H.S., McMahon, T.A., 1993. Assessing the adequacy of catchment streamflow yield estimates. Australian Journal of Soil Research 31, 665–680.

Demargne, J., Limin, W., Regonda, S. K., Brown, J. D., Lee, H., Minxue, H., Seo, D.-J., Hartman, R., Herr, H. D., Fresch, M., Schaake, J., and Zhu, Y. (2014). The science of NOAA's operational hydrologic ensemble forecast service. *American Meteorological Society*, 95(1), 79–98. https://doi.org/10.1175/BAMS-D-12-00081.1

---

## Author Comment (AC2)

Hydrol. Earth Syst. Sci. Discuss., referee comment RC1
https://doi.org/10.5194/hess-2022-72-RC1, 2022

[Figure]

**Comment on hess-2022-72**

Anonymous Referee #2

Referee comment on "Development of a national 7-day ensemble streamflow forecasting service for Australia" by Hapu Hapuarachchi et al., Hydrol. Earth Syst. Sci. Discuss., https://doi.org/10.5194/hess-2022-72-RC2, 2022

**Major comments**

Error correction vs consistency. The application of ERRIs is quite impressive in terms of taken care of the errors and producing the best reliable forecast estimate. However, I am a bit concerned about the methodology in an operational setting. You state that observed discharge is used if available, and if not the post-processed streamflow is used instead. Is there not a risk that the forecast becomes jumpy if it is initialised differently from one forecast to the other? How is this information relayed to the forecaster, and how can they take this into account when taking decisions?

*Author Response: We thank the reviewer for commending the use of ERRIS for error correction and understand the concern of its use in operational setting. Note that we do not add any noise to the simulation. The ERRIS model initial state is carried forward, and therefore the simulation and the forecast are smooth. As in the example given below, every day, we initiate the model run from 4 days prior to the forecast time. This allows us to capture most recent observed data. The model runs with observed data for 4 days and then smoothly switch to the forecast mode. If the observed data is missing, the impact of ERRIS gradually declines (switch to forecast mode) with lead-time and the corrected hydrograph overlaps with the raw simulated flow. Therefore, model simulation to forecast mode is smooth. Sometimes if poor quality observed data is ingested to the model, we find jumpiness in the forecasts. If this occurs, the model is taken temporarily out of the service, and the user community is notified through the website. The Bureau has a dedicated monitoring team to do this. To address the reviewer's question, we can elaborate on this issue in Section 6 and the last dot point of Section 5.4.*

[Figure]

Evaluation and calibration of the ensemble forecasts. Maybe I am missing something in the methodology, but it is not clear to me exactly how the optimal ensemble forecast is derived. In Section 2.3 you describe something that sounds more like a resampling from the available data than actually expanding the ensemble size (see specific comment). Later, it is mentioned that CHyPP generates 400 bias corrected forecasts. The calibration of forecast is mentioned but since no closer description of the method is given it is not clear to me how the optimal ensemble size is achieved. I suggest the authors to be clearer on these points.

*Author Response: We thank the reviewer for the comment. In section 2.3 we describe the methodology for deciding the optimal ensemble size and the results are presented in Section 4.1. As the reviewer correctly pointed out, we generated 1000 rainfall ensemble members and resampled them to find the optimal ensemble size. Alternatively, one could independently generate a different number of ensemble members and analyse them. We will elaborate the Section 2.3 to make it clear for the reader.*

Acceptance criteria. You mention here skill criterion for releasing forecasts to the public, but would not the value of the forecast be a more informed measure? In areas with high risk, even a not so skilful forecast can still be very useful.

*Author Response: We agree with the reviewer's comment, and a similar opinion was expressed by the other reviewer. The forecast skill criteria are one set of measures for selecting a forecasting location for the service. As the reviewer suggested, we consult our stakeholders and identify forecast locations critical for their decision making and add to the service. Sometimes, the forecast locations with poor skill are only made available to registered users (accessed with a username and password). This is to reduce possible miscommunication with the public and to keep the reputation of the service. To address reviewer's question, we will elaborate Sections 4.5 and 5.4.*

Minor comments

- You state that the forecasters need information on the longest possible lead time, but I would argue it depends on the action needed.

*Author Response: Yes, we agree. Through stakeholder consultation we found that they need forecast information from hours-to-days-to-months-to-years-to-decades depending on their planning and management needs. We will refine the text accordingly.*

- Reference for EFAS is missing

*Author Response: We will include EFAS in revised manuscript.*

- L94-95. This sentence could be split to increase readability

*Author Response: Yes, we will split the sentence to increase readability.*

- You start here by describing how you created the area-averaged rainfall, but I miss some information on the size of these sub-catchments. I would suggest at least introduce the hydrological modelling concept to better understand why this step is necessary.

*Author Response: We agree with the reviewer and will elaborate the hydrological modelling concept in the paper.*

- L129-136, Table 1. The description of the Super-ensemble is a bit confusing to me. When you say concatenate, I assume you mean that the ensembles are added to create a larger ensemble. I might use merge here, since concatenate to me suggests they are stitched together in time. Also, how do you create the hourly temporal resolution from the 3-hourly. There might be some feature in CHyPP method, but it is not clear

*Author Response: We agree that 'merge, is a better word than 'concatenate'. We will change the text accordingly. 3-hourly rainfall data are disaggregated to hourly using linear interpolation. Kindly refer our paper, Bennett et al., 2016 (*doi:10.1016/j.envsoft.2015.11.006*), which shows that even converting daily totals to hourly in this way produces plausible rainfall-runoff model outputs.*

- Here you describe how sub catchments are created, but I still miss information on the typical sizes. I would recommend a table or figure to show the distribution of sub basin sizes to put it into context with the resolution of the NWP models.

*Author Response: Agree with the comment. In sub-Section 2.2.1 we present typical size of sub-areas. In our modelling approach, the smallest unit is a sub-area. A collection of sub-areas makes a sub-catchment. We can elaborate the sub-catchment and sub-area sizes in the sub-*

*section. If needed, we can provide a figure showing catchment area (x-axis) and number of sub-catchments (y-axis).*

- In the evaluation framework you use the terms validation of the calibration but forecast verification. I think the term validation is good, but the term verification is very often used a bit misleading in meteorology. A forecast cannot in principle be verified since there is no absolute truth, and we are not looking for the absolute truth. We are looking for a forecast that can pass certain criteria, so the term benchmarking is to me a better term to use.

*Author Response: We agree with the reviewer that a forecast cannot in principle be verified since there is no absolute truth. This is a very literal interpretation of the term 'verification' - which relates to truth - sometimes we have seen this argument made by philosophers of science who argue models can only be 'validated' not 'verified'. However, the words do not have fixed meanings as given in dictionaries. In streamflow forecasting (and forecasting more generally) the term 'verification' is widely used (Kunnath-Poovakka and Eldho, 2007; Anctil and Ramos, 2019; Wu et al., 2020) to describe what is presented in our paper. We could change it to 'benchmarking' to satisfy the reviewer, but then our target audience - fellow forecasters - would not be clear on what we mean. Our preference is to keep the existing word 'verification' in the manuscript.*

- Section 2.3 is interesting. Normally this is not how you determine the optimal ensemble size. If I understand correctly your method you are sampling randomly from the hindcast period, thus choosing forecasts from a random starting date. The forecast skill is however very varying from time to time, so I am not sure that it is the best way of deciding the optimal skill. Would it not be better to dress the ensembles to create more members for each forecast time, than reducing the number of ensembles taking the whole hindcast period into consideration?

*Author Response: We thank the reviewer for the comment. In Section 2.3 we describe the methodology followed for deciding the optimal ensemble size and results are presented in Section 4.1. As the reviewer correctly pointed out, we generated 1000 rainfall ensemble members and resampled it to find the optimal ensemble size. Alternatively, one could independently generate different number of ensemble members and analyse them. We wish to ensure that the forecasts are true ensembles - i.e. each ensemble member can be summed across time to produce reliable forecasts of accumulations (e.g. 7-day streamflow totals). 'Ensemble dressing' methods that simply add noise to a given lead time are not suitable for this type of calculation. We will elaborate on this in Section 2.3 to make it clear for the reader.*

- Here you mention hourly forecasts from ACCESS-GE2 and ECMWF, in table 1 you mentioned 3-hourly forecasts?

*Author Response: We generate hourly rainfall forecasts after calibration using CHyPP at sub-area scale. However, the raw NWP data are 3-hourly. We acknowledge reviewer's point and will modify the text to accommodate it.*

- What is the reason for averaging over 24h before making the skill assessment? Is that not blurring the skill assessment? You will have better results, but you might miss some important information for example on timing errors in the forecast. I would suggest to also look at 3 or 6-hourly scores to see how they compare with the daily forecasts.

*Author Response: A good question. We have tested hourly and 3-hourly skill of the forecasts. However, for most of the forecast locations, it is noisy and hard to summarise the skill of the forecasts. Please find below an example of mean absolute error skill score for hourly (fig 1 – left) and daily (fig 2 – right) data for a selected location.*
*The daily forecast skill plots provide a consistent message, and they assist the users to make decisions with the forecasts. As expected, the daily forecast skill (generally) declines with lead-time. Stakeholders we communicated with prefer daily skill plots, and so these are provided to the public via the website as reference (expected skill). For consistency with the service, here we presented daily skill plots.*

[Figure]

- This is a personal preference, but I would suggest to change the order of chapter 2 and 3.

*Author Response: We are happy to accommodate this request. However, we prefer the methodology first so that the reader is clear on what data is necessary to trial the methodology.*

- You use NSE here as a metric, but it is nowadays the standard to use Kling-Gupta Efficiency.

*Author Response: We agree that KGE is commonly used in the literature nowadays, but NSE is also still very common. Among the operational forecasting community, NSE is more widely used and understood by the Bureau's stakeholders (as opposed to in the academic literature), which is why we have used it here for this service.*

- Section 2.5.5, I would suggest to merge this with the description of CRPS. I would suggest to always use CRPSS since it standardises the values automatically. CRPS(S) is

very sensitive to bias, therefore it does makes sense to decompose it into its components, or at least show also the bias alongside CRPSS

*Author Response: We agree with the reviewer, and we will merge 2.5.5 with 2.5.3.*

▪ You mention here a threshold value of 0.6. is there any particular reason this is used?

*Author Response: Similar question was asked by Reviewer 1. We adopted NSE of 0.6 from Chew and McMahon (1993) in consultation with the stakeholders. We will elaborate this section and will include the reference.*

▪ In section 4.2 you discuss the effect of calibration on the bias of the forecasts. That is all good, but I would like to see how the spread is affected by the calibration.

*Author Response: We show that calibration improves the spread (reliability) of the forecasts dramatically in figure 5.*

▪ In the same section you also show the relative CRPS of the rainfall. I would suggest to instead show the CRPSS here as a measure of skill, alternatively other scores which are more targeted towards the skill of precipitation.

*Author Response: We thank the reviewer for the comment. Our focus here is to understand the error in rainfall to explore the uncertainty contributing to the streamflow forecasts. Therefore, we thought of presenting the error (relative CRPS), not the skill. We present the streamflow forecast skill because it is useful for users and managers to make decisions. We acknowledge the reviewer's comments and agree that rainfall skill assessment is important for comparing different rainfall forecast products and we will make a note for our future research.*

▪ In section 4.3 you show the effect of error correction on the streamflow, and it is clear that removing the bias improves the forecast. What is not clear to me is if the calibration of forecasts is applied as well?

*Author Response: Thanks for the observation. Calibration is used in Section 4.2 for rainfall and not applied to streamflow. Instead, we use error modelling (ERRIS) to reduce hydrological model errors and quantify uncertainty in hydrological processes. We do not apply a calibration to streamflow, as we wish to ensure that the forecasts are true ensembles - i.e. each ensemble member can be summed across time to produce reliable forecasts of accumulations (e.g. 7-day streamflow totals). Calibration must be applied at discrete lead times, and reassembling temporal properties (e.g., with the Schaake Shuffle) is more difficult for streamflow than for rainfall. We will elaborate this section and make it clear to the reader.*

▪ The acceptance criteria of 0.6 of NSE seems to me a bit contrived. All values above zero carries some values, so it would still be useful for the users?

*Author Response: We thank the reviewer for this question. Similar statement was also made by the other reviewer. We adopted NSE of 0.6 from Chew and McMahon (1993) in consultation*

*with the stakeholders. We will elaborate this section and will include reference. We agree that additional sites where forecast are not ''scientifically acceptable'' still bring benefit to the user communities. We have responded to a similar suggestion above under the reviewer's major comments.*

- L507-514. You discuss there the value of the calibration and I agree that the method is most likely very beneficial to the users, but in the acceptance criteria you did not weigh in the users perspective (value). To be consistent I would suggest to actually add that to the acceptance criteria

*Author Response: We agree with the reviewer and will elaborate this section to cover users' perspective.*

- In the same section you mention the fact that the calibration worsen CRPS(S) for longer lead times but you do not give an explanation to this behaviour. Could you say something about that?

*Author Response: We agree with this comment. There is some explanation given L384-386 though. This is an impact from the calibration. The relative error at shorter lead times is low but it is slightly higher than raw rainfall at long lead times. However, the reliability is significantly high at longer lead-times. Normally, calibrated rainfall values are closer to climatology values at long lead times, and there is a trade-off between sharpness and reliability. We will elaborate this section and make this clear to the reader.*

- Section 5.2 I am a bit confused why you have this section. It is names uncertainties in forecast, but you almost only talk about the uncertainties in observations. I do not see the real relevance of this discussion with regards to this paper? I would suggest reducing this bit, or at least not into so much details regarding observations.

*Author Response: We thank the reviewer for the observation and suggestions. We agree, and will reduce this discussion, in particular the text related to observations.*

- Section 5.3. I really like this section and the very important discussion of the complexity of correcting forecast errors. It should also be mentioned that data assimilation has a potentially negative effect for hydrology since the water budget is compromised, which in turn can lead to long term biases in variables such as soil moisture runoff and discharge.

*Author Response: We acknowledge and agree with the reviewer's comments. We will modify this section as suggested.*

- Section 5.4 This list of challenges is good, but can you state which of these are specifically important for Australia?

*Author Response: We thank the reviewer for this suggestion. We will elaborate this section and make it more relevant to Australian context.*

- Section 6. I do not understand why this section comes here, this should have been presented at the beginning of the paper. Am I to understand that the CHyPP model "generates" 400 ensemble members form ECMWF's 51? I would need more detail or at least a very good reference to this method to understand it better.

*Author Response: We thank the reviewer for the opinion. As suggested by the other reviewer, we will swap the Section 6 and Section 5. As discussed in the results, we plan to generate 200 ensemble members from each ACCESS-GE2 and ECMWF (total 400) using CHyPP (Robertson et al., 2013). However, at the time of the operational release of the service, ACCESS-GE product was not operationally available. We tested and decided to generate 400 ensemble members from ECMWF alone in the operational system (Fig. 12) until ACCESS-GE is operationally available. The current operational system runs with these settings and the incorporation of ACCESS-GE is planned for the near future. We will elaborate this section and make it more clear to the reader.*

**References**

Anctil, F., & Ramos, M.-H. (2019). Verification metrics for hydrological ensemble forecasting. In Duan, F. Pappenberger, A. Wood, H. L. Cloke, & J. C. Schaake (Eds.), Handbook of Hydrometeorological Ensemble Forecasting. Berlin: Springer. https://doi.org/10.1007/ 978-3-642-39925-1_3

Bennett, J. C., Robertson, D. E., Ward, P. G. D., Hapuarachchi, H. A. P., Wang, Q. J.: Calibrating hourly rainfall-runoff models with daily forcings for streamflow forecasting applications in meso-scale catchments. Environmental Modelling & Software 76: 20-36. doi:10.1016/j.envsoft.2015.11.006, 2016.

Brown, J. D., Wub, L., He, M., Regonda, S., Lee, H., and Seo, D.J. (2014). Verification of temperature, precipitation, and streamflow forecasts from the NOAA/NWS Hydrologic Ensemble Forecast Service (HEFS):1. Experimental design and forcing verification, http://dx.doi.org/10.1016/j.jhydrol.2014.05.028, J. Hydrol. 519, 2869-89.

Chiew, F.H.S., McMahon, T.A., 1993. Assessing the adequacy of catchment streamflow yield estimates. Australian Journal of Soil Research 31, 665–680.

Kunnath-Poovakka, A. and Eldho, T. I.: A comparative study of conceptual rainfall-runoff models GR4J, AWBM and Sacramento at catchments in the upper Godavari river basin, India, J. Earth Syst. Sci., doi:10.1007/s12040-018-1055-8, 2019.

Robertson, D. E., Shrestha, D. L. and Wang, Q. J.: Post-processing rainfall forecasts from numerical weather prediction models for short-term streamflow forecasting, Hydrol. Earth Syst. Sci., doi:10.5194/hess-17-3587-2013, 2013.

Wu, W., Emerton, R., Duan, Q., Wood, A. W., Wetterhall, F. and Robertson, D. E.: Ensemble flood forecasting: Current status and future opportunities, WIREs Water, doi:10.1002/wat2.1432, 2020.

---

## Author Response (AR1)

Hydrol. Earth Syst. Sci. Discuss., referee comment RC1
https://doi.org/10.5194/hess-2022-72-RC1, 2022

[Figure]

**Comment on hess-2022-72**

Anonymous Referee #1
* * *
Referee comment on "Development of a national 7-day ensemble streamflow forecasting service for Australia" by Hapu Hapuarachchi et al., Hydrol. Earth Syst. Sci. Discuss., https://doi.org/10.5194/hess-2022-72-RC1, 2022
* * *
**Overview**

**(Note: All line numbers, figures, tables and sections refer to the clean copy of the manuscript)**

This is a detailed descriptive article on the methodology followed to set up an Australian ensemble streamflow service. I commend the authors on the clear description and succinct summary of what I imagine was a very large project. I believe the submission would be of interest to readers of HESS, particularly due to the value of sharing the development of operational systems with the academic community.

*Author response: We thank the reviewer for commendation and acknowledgement.*

The paper is understandable heavily focused on Australia. I have a couple of suggestions which would help make this work relevant to a wider audience. Firstly, I suggest that more context is given to help the reader understand the hydro-climatic context that the model is being validated over for example by including some maps instead of / alongside the box blots and table summaries (further comments on this are detailed below). Secondly, I would like to see more discussion of how the development of this service in Australia builds on, and moves forward, the development of ensemble streamflow services around the world. At present the work is situated in the Australia context and the reader is given limited insight into what is novel or new about this work or why a particular approach is suitable for Australia but may not have been used elsewhere. A wider review of existing literature would help support this.

*Author Response: We thank the reviewer for their suggestions. We have accommodated the reviewer's suggestions:*

- *We elaborated the introduction section of the manuscript to give greater context to different hydroclimatic regions of Australia (lines 69-85). As suggested by the reviewer, we introduced a new figure (Fig. 13) in Section 6.1. It presents the rainfall distribution across Australia.*
- *We elaborated the discussion section (Section 6) to give greater insight to what is unique and novel about this research and why it is suitable for Australia.*

From a technical perspective the work appears sound, an assessment of the strengths and limitations on the underlying data is made and a series of established verification metrics applied. The methodological steps are clearly documented throughout.

*Author Response: Thanks for the acknowledgement, we highly appreciate it.*

From an open data perspective there is no indication of the source of quality of the observed rainfall and flow data.

*Author response: We collected rainfall and discharge data from the Australian Bureau of Meteorology databases. This data is partially quality checked.*
*The collected data goes through a comprehensive quality checking using a semi-automated workflow by visualising streamflow and nearby rainfall station data side by side. This allows the modeller to identify the connection between rainfall and streamflow (i.e., there should be a high rainfall event for high discharge). This approach assists the modeller to confidently make necessary corrections to the observed data. Then the quality checked data are visually checked (plots) for further quality assurance. The corrections/modifications made to the original data are recorded (a data file) so that all the users of this dataset are aware of them. We elaborated this process in Section 3 (Lines:323-331).*
* * *
The rainfall and streamflow observations go through a comprehensive quality checking using a semi-automated workflow by visualising streamflow and nearby rainfall station data side by side. This allows the modeller to identify the connection between rainfall and streamflow (i.e., there should be a high rainfall event for high discharge). This approach assists the modeller to confidently make necessary corrections to the observed data. Rainfall data are checked for extreme values, by comparing with data from different sources, and then removing any suspicious values. Streamflow is further checked for rating issues, the rate of change, and continuous zero values because in some locations, missing values are replaced with zeros. Then the quality checked data are visually checked (plots) for further quality assurance. The corrections/modifications made to the original data are recorded (a data file) for future reference by other users of the dataset.
* * *
My main technical concerns come from the representation of extremes within the skill assessment. L66-85 sets the context of hydrological extremes in Australia and identifies both floods and droughts as particular water management challenges. The representation of high

and low flows in forecast systems leads to different challenges at different parts of the flow regime yet the discussion around model assessment does not address this as you use evaluation metrics across the full flow regime, it is well documented that it is much easier to model non-extreme flows. Is there also a need to consider the skill of the forecast system in identifying events that cross a high / low threshold as it is during these events that the system will have more operational value and your results may be skewed depending on characteristic of individual catchments. I appreciate the system is already operational and it may not be appropriate to add this to this paper, but it would be helpful to acknowledge this limitation and maybe identify it as a future research area.

*Author Response: We very much appreciate the reviewer's concerns regarding extreme events, and its performance evaluation using metrics presented in Section 2.5. We certainly agree to acknowledge this limitation and identify it as a future research area.*
*The Bureau provides more specific and specialised services for flood and drought forecasts. Therefore, if a flood is current for a location, the users of this service are redirected to Bureau's operational flood forecasting and warning website (please see a screenshot shown below – yellow banner). Therefore, we deliberately avoided discussing extreme events in this paper. However, we appreciate reviewer's comments and agree to acknowledge this limitation and identify it as a future research area.*

[Figure]

**Specific comments on the text and figures**

L56 – 65 – it is unclear to me what this paragraph on continental and global scale models adds to the paper. Could you integrate this in the context of developing a streamflow model for Australia e.g. what lessons did you learn from the existing global models?

*Author Response: Thanks for the suggestions. We have elaborated this para in particular relevance to Australia (Lines:66-68).*

> The past research demonstrates that ensemble streamflow predictions at different temporal scale is possible, but the skills vary from one geographical location to another. These findings give us greater confidence for the development of an operational ensemble streamflow forecasting service for Australia.

L98 (preprint manuscript) – do you know of other examples of "hybrid dynamical-statistical streamflow forecasting systems" or similar set ups. It would be helpful here to identify if there is anything unique about the Australian system compared to other operational systems in other countries.

*Author Response: We acknowledge reviewer's comments. Further information on other hybrid forecast systems and the uniqueness about the Australian system is discussed in Section 2 (lines 101-109).*

> This setup has several key benefits: it makes the system highly modular, allowing new models (e.g., new NWPs) to be substituted into the system without the need to revise other components (e.g., the hydrological model). Second, it means that more appropriate techniques can be applied to estimate forecast uncertainty in each case: for example, error models are better able to handle the strong autocorrelation in streamflow than statistical calibration methods typically applied to rainfall forecasts. There is only a few streamflow forecast systems around the world use the hybrid technology. The Hydrologic Ensemble Forecast System (HEFS, Demargne et al., 2014) is a hybrid forecast system. It applies a calibration to rainfall, and an error model. However, in an operational setting, it uses 'in-the-loop' flood forecasters to manually do data assimilation, which may impede the ability to produce reliable ensemble forecasts. The European Flood Awareness System (EFAS) and Global Flood Awareness System (GloFAS) use dynamical models only.

Table 3 – for those not familiar with Australian climatology it would be helpful to show some of the info in this table graphically e.g. could you include a map of mean annual rainfall distribution (or anther representative variable) across Australia, it's hard to fathom this from the table, especially as the number of catchments in each drainage division are quite varied. Other information that might be interesting is an indication of the catchment response time, are you looking at steep flashy catchments or slowly responding catchments. Later on you mention ephemeral rivers as a reason for lower forecast skill, again is there a particular region where they are more common? This type of characteristics information would help readers compare your approach to approaches taken in other countries and understand potential spatial variations in your model skill.

*Author Response: We acknowledge the reviewer's suggestions. We added a spatial map (Fig. 13) to the section 6.1 showing the forecast skill (CRPSS%) for lead-days one and three for 283 forecast locations across Australia with a mean annual rainfall in the background. This would help the readers to link catchment characteristics with the forecast skill. We also modified the text (a few locations) in the section to make it clear to the reader.*

Fig 7 – the caption and x axis label for fig 7b are inconsistent

*Author Response: We acknowledge the error and have fixed it.*

Section 4.5 Acceptance Criteria - How did you specify the 0.6 NSE threshold? Was this in conjunction with user requirements or based on existing published thresholds? Do you have any indication of the acceptable forecast skill for users? I find it interesting that there were additional sites when the forecast skill wasn't 'scientifically acceptable' yet users still wanted to receive this information. How have you addressed presenting forecast skill in the user interface? Also see my comments above re: the skill for different parts of the flow regime, did you incorporate this in any way?

*Author Response: We adopted NSE of 0.6 from Chew and McMahon (1993) in consultation with the stakeholders. We provided the reference and modified the text in section 4.5 to make it clear to the reader (line 487-500). We agree that additional sites where forecast are not "scientifically acceptable" still bring benefit to the user communities. We have responded to a similar suggestion above under the reviewer's major comments.*

*The forecast skill criteria are one set of measures for selecting a forecasting location for the service. We consulted our stakeholders and identified forecast locations critical for their decision making and added to the service. Sometimes, the forecast locations with poor skill are only available to registered users. This is to reduce possible miscommunication with the public and to keep the reputation of the service. Please find below a sample plot of forecast skill for a forecast location (with good skill) that is available for users via the service website. A description of how to read this plot is also given on the website.*
*(http://www.bom.gov.au/water/7daystreamflow/#panel=advanced).*

However, we relaxed the acceptance criteria for the locations with social and economic significance in consultation with the stakeholders. The first criterion is that the Nash Sutcliffe Efficiency (NSE) of simulated streamflow is 0.6 or greater (Chew and McMahon, 1993) for a forecast location in the model validation (see section 2.2.3). This requirement was adopted in consultation with the stakeholders to maintain the service standard. It ensures the hydrological model is robust and produces acceptable results with observed data. If the first criterion is met, then forecast skill (CRPSS), with reference to climatology, should be consecutively positive up to three days lead-time (Fig. 11). We calculate model performance metrics for each forecast location, and only if the criteria for a forecast location are satisfied, it is added to the public service. Poor quality forecasts possibly lead to miscommunicating the flow conditions with the public. It impacts the reputation of the service and the organization. If only the first criterion is satisfied, we consider releasing the forecasts only to registered users based on stakeholder requirements and the social and economic importance of forecasts at the location. Some water agencies use their own tools to generate streamflow forecasts. Therefore, consistently maintaining the forecast quality is important for a national operational service. If the first criterion is not met, then the forecast location is unsuitable for the service, and further revision of the model is required. We modelled 283 potential forecast locations in 100 catchments for the current service, and of these, 209 forecast locations in 99 catchments pass the acceptance criteria and are released to the public.

[Figure]

Section 5.1 goes on to discuss some reasons for variability in forecast skill, could you show the forecast skill spatially on a map and any links to catchment/meteorological forecast characteristics? Again the table display in Table 4 is difficult to interpret due to the number of forecast locations lumped into each jurisdiction.

*Author Response: We acknowledge the reviewer's suggestion, and we included a figure (Fig. 12) showing forecast skills for different locations across Australia (Lines 535-560).*

Section 5 (Now 6) is interesting and raises established challenges of operational streamflow forecasting however it lacks integration with the rest of the paper. Possibly this could be improved with incorporation of wider literature on development of streamflow forecasting systems mentioned above. I also suggest it is moved after section 6 (Now 5) so that it links to the summary and conclusions section.

*Author Response: We acknowledge the reviewer's suggestions. We elaborated Section 6 (Now 5) to integrate more closely with the results and Australian relevance as presented in previous sections. Also, we changed the order of Section 5 and 6 (Now 6 and 5 respectively) as suggested by the reviewer.*

**Reference**

Chiew, F.H.S., McMahon, T.A., 1993. Assessing the adequacy of catchment streamflow yield estimates. Australian Journal of Soil Research 31, 665–680.

Demargne, J., Limin, W., Regonda, S. K., Brown, J. D., Lee, H., Minxue, H., Seo, D.-J., Hartman, R., Herr, H. D., Fresch, M., Schaake, J., and Zhu, Y. (2014). The science of NOAA's operational

hydrologic ensemble forecast service. *American Meteorological Society*, 95(1), 79–98. https://doi.org/10.1175/BAMS-D-12-00081.1

**Comment on hess-2022-72**
Anonymous Referee #2

Referee comment on "Development of a national 7-day ensemble streamflow forecasting service for Australia" by Hapu Hapuarachchi et al., Hydrol. Earth Syst. Sci. Discuss., https://doi.org/10.5194/hess-2022-72-RC2, 2022

**(Note: All line numbers, figures, tables and sections refer to the clean copy of the manuscript)**

**Major comments**

Error correction vs consistency. The application of ERRIs is quite impressive in terms of taken care of the errors and producing the best reliable forecast estimate. However, I am a bit concerned about the methodology in an operational setting. You state that observed discharge is used if available, and if not the post-processed streamflow is used instead. Is there not a risk that the forecast becomes jumpy if it is initialised differently from one forecast to the other? How is this information relayed to the forecaster, and how can they take this into account when taking decisions?

*Author Response: We thank the reviewer for commending the use of ERRIS for error correction and understand the concern of its use in operational setting. Note that we do not add any noise to the simulation. The ERRIS model initial state is carried forward, and therefore the simulation and the forecast are smooth. As in the example given below, every day, we initiate the model run from 4 days prior to the forecast time. This allows us to capture most recent observed data. The model runs with observed data for 4 days and then smoothly switch to the forecast mode. If the observed data is missing, the impact of ERRIS gradually declines (switch to forecast mode) with lead-time and the corrected hydrograph overlaps with the raw simulated flow. Therefore, model simulation to forecast mode is smooth. Sometimes if poor quality observed data is ingested to the model, we find jumpiness in the forecasts. If this occurs, the model is taken temporarily out of the service, and the user community is notified through the website. The Bureau has a dedicated monitoring team to do this. To address the reviewer's concerns, we discussed further details in the last dot point of Section 6.4 (Lines 678-681).*

Particularly, if poor quality observed data is ingested to a model, we find discontinuity in the output when transition from simulation to forecast due to the instability of the ERRIS model. If this occurs, the model is taken temporarily out of the service by the monitoring team, and the user community is notified through the website.

[Figure]

Ovens River at Wangaratta (ID: 403242A)
Forecast for 22 May 2022 to 29 May 2022 (09:00 AEST)

IDA7Y4031B, Generated: 2022-05-22T00:57 UTC (v1.3.1)

©Commonwealth of Australia 2022, Australian Bureau of Meteorology

Evaluation and calibration of the ensemble forecasts. Maybe I am missing something in the methodology, but it is not clear to me exactly how the optimal ensemble forecast is derived. In Section 2.3 you describe something that sounds more like a resampling from the available data than actually expanding the ensemble size (see specific comment). Later, it is mentioned that CHyPP generates 400 bias corrected forecasts. The calibration of forecast is mentioned but since no closer description of the method is given it is not clear to me how the optimal ensemble size is achieved. I suggest the authors to be clearer on these points.

*Author Response: We thank the reviewer for the comment. We provided more information about the resampling and optimum ensemble size selection process in the section 2.3 (lines 210-215) and section 4.1 (lines 361-364).*

Alternatively, different number of ensemble member ($m$) samples can be generated independently and analyse them. However, it needs extensive computational resources that is unavailable to us. Another method is to dress the ensembles to create more members for each forecast time. However, we want to ensure that the forecasts are true ensembles - i.e., each ensemble member can be summed across time to produce reliable forecasts of accumulations (e.g., 7-day streamflow totals). 'Ensemble dressing' methods that simply add noise to a given lead time are not suitable for this type of calculation.

Merging calibrated ACCESS-GE2, and ECMWF products will not negatively impact on the forecast accuracy (see section 4.3) since they show similar forecast accuracy for the same sample size. The recommendation of 200 calibrated ensemble members from each of the two rainfall forecast products is drawn considering the results shown in Figure 3, particularly to meet the operational computational efficiency and resources availability at the BoM.

Acceptance criteria. You mention here skill criterion for releasing forecasts to the public, but would not the value of the forecast be a more informed measure? In areas with high risk, even a not so skilful forecast can still be very useful.

*Author Response: We agree with the reviewer's comment, and a similar opinion was expressed by the other reviewer. The forecast skill criteria are one set of measures for selecting a forecasting location for the service. As the reviewer suggested, we regularly consult our stakeholders and identify forecast locations critical for their decision making and add to the service. Sometimes, the forecast locations with poor skill are only made available to registered users (accessed with a username and password). This is to reduce possible miscommunication with the public and to keep the reputation of the service. We modified the Sections 4.5 to address reviewer's comments (lines 492-498).*

We calculate model performance metrics for each forecast location, and only if the criteria for a forecast location are satisfied, it is added to the public service. Poor quality forecasts possibly lead to miscommunicating the flow conditions with the public. It impacts the reputation of the service and the organization. If only the first criterion is satisfied, we consider releasing the forecasts only to registered users based on stakeholder requirements and the social and economic importance of forecasts at the location. Some water agencies use their own tools to generate streamflow forecasts. Therefore, consistently maintaining the forecast quality is important for a national operational service.

Minor comments

- You state that the forecasters need information on the longest possible lead time, but I would argue it depends on the action needed.

*Author Response: Yes, we agree. Through stakeholder consultation we found that they need forecast information from hours-to-days-to-months-to-years-to-decades depending on their planning and management needs. We modified the text from line 43 to 44 in the introduction.*

Water and flood managers need accurate streamflow forecast information with a useful lead time to make optimal water management decisions. The useful lead-time can be hours to years depending on the type of application and actions required.

- Reference for EFAS is missing

*Author Response: We provided the reference (line 59).*

Smith, P. J. *et al.* (2016) 'Chapter 11 - On the Operational Implementation of the European Flood Awareness System (EFAS)', in Adams, T. E. and Pagano, T. C. B. T.-F. F. (eds). Boston: Academic Press, pp. 313–348. doi: https://doi.org/10.1016/B978-0-12-801884-2.00011-6.

- L94-95 (Preprint manuscript). This sentence could be split to increase readability

*Author Response: We rewrote the sentence (line 97-98)*

> The adopted hybrid dynamical-statistical streamflow forecasting method consists of several components. It includes NWP calibration, hydrological modelling and hydrological error modelling and we firstly introduce the components of the system.

- You start here by describing how you created the area-averaged rainfall, but I miss some information on the size of these sub-catchments. I would suggest at least introduce the hydrological modelling concept to better understand why this step is necessary.

*Author Response: We agree with the reviewer and provided further information in section 2.1 (lines 123-26).*

> A catchment is delineated to sub-catchments and finer sub-areas to represent a semi-distributed model structure. A hydrological model is applied to each sub-area with average areal rainfall (see section 2.2.1). The average areal rainfall of each sub-area per each ensemble member is calculated by taking the area-weighted average of gridded forecast rainfall for all grid cells intersecting the sub-area.

- L129-136, Table 1. The description of the Super-ensemble is a bit confusing to me. When you say concatenate, I assume you mean that the ensembles are added to create a larger ensemble. I might use merge here, since concatenate to me suggests they are stitched together in time. Also, how do you create the hourly temporal resolution from the 3-hourly. There might be some feature in CHyPP method, but it is not clear

*Author Response: We agree that 'merge, is a better word than 'concatenate'. We changed the text accordingly. 3-hourly rainfall data are disaggregated to hourly using linear interpolation. We modified the text in section 2.1 (lines 139-141) as suggested by the reviewer.*

> The 3 and 6 hourly NWP data (Table 1) are disaggregated to hourly using linear interpolation within CHyPP. Bennett et al. (2016) showed that even converting daily rainfall totals to hourly using linear interpolation produces plausible rainfall-runoff model outputs..

- Here you describe how sub catchments are created, but I still miss information on the typical sizes. I would recommend a table or figure to show the distribution of sub basin sizes to put it into context with the resolution of the NWP models.

*Author Response: Agree with the comment. We added more information about the sub-area sizes in the section 2.2.1 (lines 164-166)*

For the 100 catchments, size of a sub-area varies from 30 to 4000 km$^2$ of which the mean and median values are 600 and 450 km$^2$ respectively. Larger sub-areas are present where the rainfall gradient is insignificant, and rainfall and water level gauge networks are sparse.

- In the evaluation framework you use the terms validation of the calibration but forecast verification. I think the term validation is good, but the term verification is very often used a bit misleading in meteorology. A forecast cannot in principle be verified since there is no absolute truth, and we are not looking for the absolute truth. We are looking for a forecast that can pass certain criteria, so the term benchmarking is to me a better term to use.

*Author Response: We agree with the reviewer that a forecast cannot in principle be verified since there is no absolute truth. This is a very literal interpretation of the term 'verification' - which relates to truth - sometimes we have seen this argument made by philosophers of science who argue models can only be 'validated' not 'verified'. However, the words do not have fixed meanings as given in dictionaries. In streamflow forecasting (and forecasting more generally) the term 'verification' is widely used (Kunnath-Poovakka and Eldho, 2007; Anctil and Ramos, 2019; Wu et al., 2020) to describe what is presented in our paper. We could change it to 'benchmarking' to satisfy the reviewer, but then our target audience - fellow forecasters - would not be clear on what we mean. Therefore, we want to keep the existing word 'verification' in the manuscript.*

- Section 2.3 is interesting. Normally this is not how you determine the optimal ensemble size. If I understand correctly your method you are sampling randomly from the hindcast period, thus choosing forecasts from a random starting date. The forecast skill is however very varying from time to time, so I am not sure that it is the best way of deciding the optimal skill. Would it not be better to dress the ensembles to create more members for each forecast time, than reducing the number of ensembles taking the whole hindcast period into consideration?

*Author Response: We thank the reviewer for the comment. In Section 2.3 we described the methodology followed for deciding the optimal ensemble size and results are presented in Section 4.1. As the reviewer correctly pointed out, we generated 1000 rainfall ensemble members and resampled it to find the optimal ensemble size. Alternatively, one could independently generate different number of ensemble members and analyse them. We want to ensure that the forecasts are true ensembles - i.e., each ensemble member can be summed across time to produce reliable forecasts of accumulations (e.g., 7-day streamflow totals). 'Ensemble dressing' methods that simply add noise to a given lead time are not suitable for this type of calculation. We elaborated on this in Section 2.3 (line 210-215) to make it clear for the reader.*

Alternatively, different number of ensemble member (*m*) samples can be generated independently and analyse them. However, it needs extensive computational resources that is unavailable to us. Another method is to dress the ensembles to create more members for each forecast time. However, we want to ensure that the forecasts are true ensembles - i.e., each ensemble member can be summed across time to produce reliable forecasts of accumulations (e.g., 7-day streamflow totals). 'Ensemble dressing' methods that simply add noise to a given lead time are not suitable for this type of calculation.

- Here you mention hourly forecasts from ACCESS-GE2 and ECMWF, in table 1 you mentioned 3-hourly forecasts?

*Author Response: We generate hourly rainfall forecasts after rainfall calibration using CHyPP at sub-area scale. However, the raw NWP data are 3-6 hourly. We acknowledge reviewer's point. We modified the text to make it clear to the reader (line 139-141).*

The 3 and 6 hourly NWP data (Table 1) are disaggregated to hourly using linear interpolation within CHyPP. Bennett et al. (2016) showed that even converting daily rainfall totals to hourly using linear interpolation produces plausible rainfall-runoff model outputs.

- What is the reason for averaging over 24h before making the skill assessment? Is that not blurring the skill assessment? You will have better results, but you might miss some important information for example on timing errors in the forecast. I would suggest to also look at 3 or 6-hourly scores to see how they compare with the daily forecasts.

*Author Response: A good question. We have tested hourly and 3-hourly skill of the forecasts. However, for most of the forecast locations, it is noisy and hard to summarise the skill of the forecasts. Please find below an example of mean absolute error skill score for hourly (fig 1 – left) and daily (fig 2 – right) data for a selected location.*
*The daily forecast skill plots provide a consistent message, and they assist the users to make decisions with the forecasts. As expected, the daily forecast skill (generally) declines with lead-time. Stakeholders we communicated with prefer daily skill plots, and so these are provided to the public via the website as a reference (expected skill). For consistency with the service, here we presented daily skill plots.*

[Figure]

[Figure]

- This is a personal preference, but I would suggest to change the order of chapter 2 and 3.

*Author Response: We acknowledge reviewer's comment. However, we prefer the methodology first so that the reader is clear on what data is necessary to trial the methodology.*

- You use NSE here as a metric, but it is nowadays the standard to use Kling-Gupta Efficiency.

*Author Response: We agree that KGE is commonly used in the literature nowadays. However, NSE is also still very common. Among the operational forecasting community, NSE is more widely used and well understood by the Bureau's stakeholders (as opposed to in the academic literature), which is why we have used it here for the service.*

- Section 2.5.5, I would suggest to merge this with the description of CRPS. I would suggest to always use CRPSS since it standardises the values automatically. CRPS(S) is very sensitive to bias, therefore it does makes sense to decompose it into its components, or at least show also the bias alongside CRPSS

*Author Response: We agree with the reviewer. We included CRPSS in section 2.5.3 (line 269-274) and deleted section 2.5.5.*

Skill is a measure of relative improvement of the forecast over a reference forecast. The Continuous Ranked Probability Skill Score (CRPSS) is given by:

$$CRPSS = 1 - \frac{CRPS_{forecast}}{CRPS_{reference}} \qquad (5)$$

where $CRPS_{reference}$ is the reference forecast. For this study, we use climatology as the reference forecast. Data from 1990 to 2016 are used for climatological streamflow calculation. For any given day of the year, the climatology value is the median of the period from 2 weeks before that day to 2 weeks after (i.e., 29 days) over the climatology period excluding the forecast year.

- You mention here a threshold value of 0.6. is there any particular reason this is used?

*Author Response: We acknowledge the reviewer's comment. We adopted NSE of 0.6 from Chew and McMahon (1993) in consultation with the stakeholders. We provided the reference and modified the text in section 4.5 to make it clear to the reader (line 488-489).*

- In section 4.2 you discuss the effect of calibration on the bias of the forecasts. That is all good, but I would like to see how the spread is affected by the calibration.

*Author Response: We show that calibration improves the spread (reliability) of the forecasts dramatically in figure 5.*

- In the same section you also show the relative CRPS of the rainfall. I would suggest to instead show the CRPSS here as a measure of skill, alternatively other scores which are more targeted towards the skill of precipitation.

*Author Response: We thank the reviewer for the comment. Our focus here is to understand the error in rainfall to explore the uncertainty contributing to the streamflow forecasts. Therefore, we thought of presenting the error (relative CRPS), not the skill. We present the streamflow forecast skill because it is useful for the users and managers to make decisions. We acknowledge the reviewer's comments and agree that rainfall skill assessment is important for comparing different rainfall forecast products and we will make a note for our future research.*

- In section 4.3 you show the effect of error correction on the streamflow, and it is clear that removing the bias improves the forecast. What is not clear to me is if the calibration of forecasts is applied as well?

*Author Response: Thanks for the observation. Calibration is used in Section 4.2 for rainfall and not applied to streamflow. Instead, we use error modelling (ERRIS) to reduce hydrological model errors and quantify uncertainty in hydrological processes. We do not apply a calibration to streamflow, as we wish to ensure that the forecasts are true ensembles - i.e. each ensemble member can be summed across time to produce reliable forecasts of accumulations (e.g., 7-day streamflow totals). Calibration must be applied at discrete lead times, and reassembling temporal properties (e.g., with the Schaake Shuffle) is more difficult for streamflow than for rainfall.*

- The acceptance criteria of 0.6 of NSE seems to me a bit contrived. All values above zero carries some values, so it would still be useful for the users?

*Author Response: We thank the reviewer for this question. Similar statement was also made by the other reviewer. We adopted NSE of 0.6 from Chew and McMahon (1993) in consultation with the stakeholders. We provided the reference and modified the text in section 4.5 to make it clear to the reader (line 488-489). We agree that additional sites where forecast are not ''scientifically acceptable'' still bring benefit to the user communities. We have responded to a similar suggestion above under the reviewer's major comments.*

> The first criterion is that the Nash Sutcliffe Efficiency (NSE) of simulated streamflow is 0.6 or greater (Chew and McMahon, 1993) for a forecast location in the model validation (see section 2.2.3).
>
> Chiew, F.H.S., McMahon, T.A., 1993. Assessing the adequacy of catchment streamflow yield estimates. Australian Journal of Soil Research 31, 665–680.

- L507-514 (preprint manuscript). You discuss there the value of the calibration and I agree that the method is most likely very beneficial to the users, but in the acceptance

criteria you did not weigh in the users perspective (value). To be consistent I would suggest to actually add that to the acceptance criteria

*Author Response: We agree with the reviewer and modified the text in section 4.5 (line 485-488) to make it clear to the reader.*

> In consultation with key stakeholders, we developed criteria, based on model performance and forecast skill for accepting forecast locations for the operational service. However, we relaxed the acceptance criteria for the locations with social and economic significance in consultation with the stakeholders.

- In the same section you mention the fact that the calibration worsen CRPS(S) for longer lead times but you do not give an explanation to this behaviour. Could you say something about that?

*Author Response: We agree with this comment. There is some explanation given L384-386 though. This is an impact from the calibration. The relative error at shorter lead times is low but it is slightly higher than raw rainfall at long lead times. However, the reliability is significantly high at longer lead-times. Normally, calibrated rainfall values are closer to climatology values at long lead times, and there is a trade-off between sharpness and reliability. We revised this section (4.2) to make it clear to the reader (line 407-410).*

> Normally calibrated rainfall values are closer to climatology values at long lead times giving more weight to improving the reliability. This is an inherent characteristic of the CHyPP methodology and there is a trade-off between sharpness and reliability. Further research is needed to explore on how to keep the balance between the sharpness and reliability of the calibrated forecast rainfall at long lead times.

- Section 5.2 (Now 6.2) I am a bit confused why you have this section. It is names uncertainties in forecast, but you almost only talk about the uncertainties in observations. I do not see the real relevance of this discussion with regards to this paper? I would suggest reducing this bit, or at least not into so much details regarding observations.

*Author Response: We thank the reviewer for the observation and suggestions. We revised the text in section 5.2 (line 582-588).*

Input data (observations and forecasts) and hydrological model structural uncertainties contribute to the streamflow forecast uncertainties. We try to minimise input data uncertainty by calibrating NWP rainfall forecasts using the CHyPP model and minimise the hydrologic uncertainty by applying the ERRIS error model to simulated discharge. We demonstrate that calibrated NWP rainfall forecasts improve streamflow forecast skill. Similar results are found in Canada and South America (Rogelis and Werner, 2018; Jha et al., 2018). However, uncertainties may also arise from the observed data used to calibrate parameters in the hydrologic models. The most common issues are precision of the instruments that measure the water level (stage) and rainfall, derivation of the stage-discharge relationship (rating tables), the accuracy of gauged rainfall interpolation methods (e.g., inverse-distance squared-weighted averaging), and data disaggregation methods. Measurement and rating curve uncertainties in streamflow, particularly for low and high flows, bring additional complexities in model calibration/validation and ultimately model performance (Tomkins, 2014).

Although there are many complex methods available for climate data disaggregation (Breinl and Di Baldassarre, 2019; Görner et al., 2021; Mehrotra and Singh, 1998), for simplicity, we use a simple method, linear interpolation for disaggregating daily rainfall and PE data to hourly. PE varies with the diurnal cycle and usually shows some degree of (negative) correlation with rainfall that could have been considered in the disaggregation. However, we note that the impact of rainfall uncertainty has been shown to be more significant than PE in hydrological modelling (Paturel, Servat and Vassiliadis, 1995; Guo et al., 2017). The sparseness of the rainfall observation network in much of inland Australia (particularly in the inland desert regions and in northwest Australia) remains a challenge for the development of any streamflow forecasting system.

- Section 5.3 (Now 6.3). I really like this section and the very important discussion of the complexity of correcting forecast errors. It should also be mentioned that data assimilation has a potentially negative effect for hydrology since the water budget is compromised, which in turn can lead to long term biases in variables such as soil moisture runoff and discharge.

*Author Response: We acknowledge and agree with the reviewer's comments. We modified this section as suggested (Line:615-619).*

However, implementation of a data assimilation method for probabilistic streamflow forecasting in a semi-distributed modelling setup is challenging due to: (i) the complexity in inter-dependencies of uncertainty contributing sources such as an ensemble of model forcing data, (ii) model state variables and/or model parameters, and (iii) compromise in landscape water balance which may lead to long term biases in streamflow forecasts (Moradkhani et al., 2005; Li et al., 2016).

- Section 5.4 (Now 6.4) This list of challenges is good, but can you state which of these are specifically important for Australia?

*Author Response: We thank the reviewer for this suggestion. We elaborated this section and made it more relevant to Australian context (Line:631-634; 684-686).*

> Some of these challenges relevant to Australian context for ensemble streamflow forecasting research, operational application, adoption, and benefit to the community are
>
> Length of the sub-daily rainfall records vary from one station to another – to a maximum of 50 years.
>
> Each of these challenges shares a real-world perspective and its relative importance vary across different geographical regions of Australia. It opens ongoing research and development opportunities, resulting in a greater update of ensemble streamflow forecasting for operational decision-making.

- Section 6. I do not understand why this section comes here, this should have been presented at the beginning of the paper. Am I to understand that the CHyPP model "generates" 400 ensemble members form ECMWF's 51? I would need more detail or at least a very good reference to this method to understand it better.

*Author Response: We thank the reviewer for the opinion. As suggested by the other reviewer, we will swap the Section 6 and Section 5. As discussed in the results, we planned to generate 200 ensemble members from each ACCESS-GE2 and ECMWF (total 400) using CHyPP. You may find details of the CHyPP method in Robertson et al. (2013). However, at the time of the operational release of the service, ACCESS-GE product was not operationally available. We tested and decided to generate 400 ensemble members from ECMWF alone in the operational system (Fig. 12) until ACCESS-GE is operationally available. The current operational system runs with these settings and the incorporation of ACCESS-GE is planned for the near future.*

**References**

Anctil, F., & Ramos, M.-H. (2019). Verification metrics for hydrological ensemble forecasting. In Duan, F. Pappenberger, A. Wood, H. L. Cloke, & J. C. Schaake (Eds.), Handbook of Hydrometeorological Ensemble Forecasting. Berlin: Springer. https://doi.org/10.1007/ 978-3-642-39925-1_3

Bennett, J. C., Robertson, D. E., Ward, P. G. D., Hapuarachchi, H. A. P., Wang, Q. J.: Calibrating hourly rainfall-runoff models with daily forcings for streamflow forecasting applications in meso-scale catchments. Environmental Modelling & Software 76: 20-36. doi:10.1016/j.envsoft.2015.11.006, 2016.

Brown, J. D., Wub, L., He, M., Regonda, S., Lee, H., and Seo, D.J. (2014). Verification of temperature, precipitation, and streamflow forecasts from the NOAA/NWS Hydrologic Ensemble Forecast Service (HEFS):1. Experimental design and forcing verification, http://dx.doi.org/10.1016/j.jhydrol.2014.05.028, J. Hydrol. 519, 2869-89.

Chiew, F.H.S., McMahon, T.A., 1993. Assessing the adequacy of catchment streamflow yield estimates. Australian Journal of Soil Research 31, 665–680.

Kunnath-Poovakka, A. and Eldho, T. I.: A comparative study of conceptual rainfall-runoff models GR4J, AWBM and Sacramento at catchments in the upper Godavari river basin, India, J. Earth Syst. Sci., doi:10.1007/s12040-018-1055-8, 2019.

Robertson, D. E., Shrestha, D. L. and Wang, Q. J.: Post-processing rainfall forecasts from numerical weather prediction models for short-term streamflow forecasting, Hydrol. Earth Syst. Sci., doi:10.5194/hess-17-3587-2013, 2013.

Wu, W., Emerton, R., Duan, Q., Wood, A. W., Wetterhall, F. and Robertson, D. E.: Ensemble flood forecasting: Current status and future opportunities, WIREs Water, doi:10.1002/wat2.1432, 2020.

---

## Author Response (AR2)

**Editor's comments and author responses**

**Comments to the author**:

Dear authors,

Thank you for your responses and the carefully revised manuscript, I am pleased to accept your paper for publication subject to minor corrections. I provide a list of corrections as follows. Please note the line numbers below are from the revised manuscript with tracked changes.

1.  Line 162-170: "delineate each catchment into sub-catchments and sub-areas", "For the 100 catchments, size of a sub-area". It is unclear what type of sub-areas they represent. Are the sub-areas like hydrological similar areas or are they simply sub-catchments as you said you used flow direction map to delineate? If the latter, please use 'sub-catchments' consistently as you mixed the two terms in the text.

    **Author response:** We acknowledge the editor's suggestion. The catchment delineation is only based on flow direction. However, we want to define three types of modelling units: catchment, sub-catchment, and sub-area. The smallest modelling unit is a sub-area to which a hydrologic model is applied. A collection of upstream sub-areas makes a sub-catchment. The hydrologic model is calibrated for each sub-catchment where all the sub-areas within a sub-catchment have the same parameters. We cannot calibrate the model to each sub-area since the observed discharge data are limited. Sub-areas are useful for representing the spatial rainfall distribution within the sub-catchment and enable stable channel routing. The precipitation and potential evapotranspiration in sub-areas within a sub-catchment are different. Therefore, the state variables and runoff in sub-areas are different.

    Kindly note that we revised the text from lines 161 to 173 (line numbers refer to track change document) to make it clear to the reader.

2.  L210-215: "Alternatively, different number of ensemble member (m) samples can be generated independently and analyse them. However, it needs extensive computational resources that is unavailable to us. Another method is to dress the ensembles to create more members for each forecast time. However, we want to ensure that the forecasts are true ensembles".

    Change 'and analyse them' to 'analysed'

    'dress the ensembles to create more members': the word 'dress' may be a jargon for the scientific community in ensemble forecasting. It needs to be explained here what you mean by 'dress' or refer to the literature that used and explained this word. Similarly, 'Ensemble dressing' appeared on the next page also needs explanation.

    **Author response:**

    Line 216: Changed the text 'and analyse them' to 'analysed'.

    Lines 217 & 220: Added references (Pagano et al., 2012; Verkade et al., 2017) that explain the ensemble dressing method.

3. Section 4.5 This new sentence "However, we relaxed the acceptance criteria for the locations with social and economic significance in consultation with the stakeholders." Should be placed closer to the last sentence of the section "On users' request, a further 17 forecast locations (including one additional catchment) with forecast skill slightly below the acceptance benchmark, are released to registered users only due to the economic and social significance of the forecasts." The new sentence also needs rephrasing to minimising repeated information.

**Author response:** As suggested by the editor, we deleted the first sentence in line 492-493 and, reworded and inserted to lines 506 & 508.

4. L643-644: However, implementation of a data assimilation method for probabilistic streamflow forecasting in a semi-distributed modelling setup is challenging.

It is not only true in a semi-distributed model, but also for any hydrological model be it a lumped, semi-lumped or distributed. So, I would change 'in a semi-distributed modelling setup' to 'using a hydrological model'.

**Author response:** We modified the text (Line 622) as suggested by the editor.

5. L705-710 "Particularly, if poor quality observed data is ingested to a model," Please state observed precipitation data, otherwise it is ambiguous and unclear what observed data you are referring here. Please consider changing the phrase 'ingested to' to perhaps 'used in'.

**Author response:** We modified the text (Lines 684-685) as suggested by the editor.

6. Section 6.4: you added "challenges relevant to Australian context", but some of the challenges are also valid for countries other than Australia. It also makes the paper very limited to the country. I would remove it or just mention within bullet's text when something is particularly relevant to Australia.

**Author response:** We modified the text (Line 637) as suggested by the editor.

References

Pagano, T. C., Shrestha, D. L., Wang, Q. J., Robertson, D., and Hapuarachchi, P.: Ensemble dressing for hydrological applications, Hydrol. Process., 27, 106–116, https://doi.org/https://doi.org/10.1002/hyp.9313, 2013.

Verkade, J. S., Brown, J. D., Davids, F., Reggiani, P., and Weerts, A. H.: Estimating predictive hydrological uncertainty by dressing deterministic and ensemble forecasts; a comparison, with application to Meuse and Rhine, J. Hydrol., 555, 257–277, https://doi.org/https://doi.org/10.1016/j.jhydrol.2017.10.024, 2017.

---

## Author Response (AR3)

**Editor's Comments**:

Dear authors,

Thank you for sending through your revised manuscript. I have one final comment as follows.

L162-165: used to delineate each catchment into sub-catchments and finer sub-areas to represent a semi-distributed model structure. A collection of upstream sub-areas makes a sub-catchment where a streamflow gauge exists at the outlet. Sub-areas are useful for representing the spatial rainfall distribution within a sub-catchment and enable stable channel routing.

It is still unclear to me how these sub-areas are defined. From your subsequent text, it seems the sub-areas are not delineated based on flow direction but divided based on the spatial rainfall distribution. Please clarify this. It might be more appropriate to say 'used to delineate each catchment into sub-catchments which was then further divided into sub-areas based on xxxxx. It therefore represents a semi-distributed model structure.'

Sincerely,

Yi He, HESS Editor

**Author response**

Dear Editor,

Thanks for your comments. We have revised the relevant section of the manuscript accordingly. Please see line numbers 161 to 164 of the track change version of the manuscript.

A nationally consistent flow direction map from the Australian Hydrological Geospatial Fabric (Geofabric) (Atkinson et al., 2008) is used to delineate each catchment into sub-catchments which is then further divided into and finer sub-areas. It therefore represents a semi-distributed model structure. A collection of upstream sub-areas makes a sub-catchment where a streamflow gauge exists at the outlet. Sub-areas are also useful for representing the spatial rainfall distribution within a sub-catchment and enable stable channel routing. The number of sub-areas varies for each catchment, depending on catchment size

165

Kind Regards,

Mohammed Bari